# Correction of dysregulated lipid metabolism normalizes gene expression in oligodendrocytes and prolongs lifespan in female poly-GA C9orf72 mice

Clinical and genetic research links altered cholesterol metabolism with ALS development and progression, yet pinpointing specific pathomechanisms remain challenging. We investigated how cholesterol dysmetabolism interacts with protein aggregation, demyelination, and neuronal loss in ALS. Bulk RNAseq transcriptomics showed decreased cholesterol biosynthesis and increased cholesterol export in ALS mouse models (GA-Nes, GA-Camk2a GA-CFP, rNLS8) and patient samples (spinal cord), suggesting an adaptive response to cholesterol overload. Consequently, we assessed the efficacy of the cholesterol-binding drug 2-hydroxypropyl-β-cyclodextrin (CD) in a fast-progressing C9orf72 ALS mouse model with extensive poly-GA expression and myelination deficits. CD treatment normalized cholesteryl ester levels, lowered neurofilament light chain levels, and prolonged lifespan in female but not male GA-Nes mice, without impacting poly-GA aggregates. Single nucleus transcriptomics indicated that CD primarily affected oligodendrocytes, significantly restored myelin gene expression, increased density of myelinated axons, inhibited the disease-associated oligodendrocyte response, and downregulated the lipid-associated genes Plin4 and ApoD. These results suggest that reducing excess free cholesterol in the CNS could be a viable ALS treatment strategy.

In amyotrophic lateral sclerosis (ALS), the degeneration of both cortical and spinal motor neurons leads to progressive paralysis and eventually respiratory failure[1,2]. All sporadic and most genetic cases show cytoplasmic aggregates of the nuclear RNA-binding protein TDP-43, found predominantly in neurons and less frequently in oligodendrocytes playing a key role in neurodegeneration and neuroinflammation[3–5]. About 5–10% of ALS cases are caused by an $(G_4C_2)_n$ repeat expansion in the first intron of *C9orf72*, which is translated by a non-canonical mechanism in all reading frames into five dipeptide repeat (DPR) proteins co-aggregating in neurons[6–10]. The most abundant form is poly-GA, which was shown to contribute to TDP-43 aggregation[11].

Secondary to neuron loss, ALS patients show demyelination in the motor cortex, corticospinal tract, and the ventral horn of the spinal cord[12,13], suggesting that oligodendrocyte dysfunction may affect disease progression. Recently, oligodendrocytes have been linked to a number of neurodegenerative diseases, including Alzheimer's disease[14–16] and ALS[17]. This is further supported by the recent identification of a shared gene expression signature in myelinating oligodendrocytes, termed disease-associated oligodendrocytes (DOLs), in mouse models of Alzheimer's disease and multiple sclerosis[14,18–21].

The myelin sheath is a lipid-rich multilayered plasma membrane extension of oligodendrocytes with a particularly high content of

✉e-mail: dieter.edbauer@dzne.de

cholesterol[22]. Myelin debris is cleared by microglia, but cholesterol cannot be fully degraded and may accumulate in cholesterol crystals or lipid droplets, leading to "foam cell" formation in primary demyelinating diseases. Effective remyelination requires cholesterol clearance and efflux by microglia[23], which uses ApoE lipoproteins as the major lipid carrier. The common ApoE4 allele is a major risk factor for Alzheimer's disease. Interestingly, ApoE4 promotes cholesterol deposition in oligodendrocytes, which can also be reversed by 2-hydroxypropyl-β-cyclodextrin (hereafter referred to as CD) treatment in cells and animals[15]. Cyclodextrins are cyclic oligosaccharides that increase the cholesterol solubility and excretion. CD is beneficial in experimental models of atherosclerosis[24] and was clinically tested in Niemann Pick disease Type C1, but side effects of intrathecal injection were not tolerated[25]. Currently, a new formulation of CD is tested clinically in Niemann Pick disease and Alzheimer's disease. Presumably through improved cholesterol clearance, cyclodextrins can also have anti-inflammatory effects[24].

Epidemiological studies reveal the complexity of cholesterol metabolism in ALS[26]. Large longitudinal studies show that ALS patients typically exhibit higher LDL cholesterol levels before onset[27] and that increased HDL cholesterol levels are associated with a reduced ALS risk[28]. Interestingly, at diagnosis, higher LDL levels are associated with slower disease progression and improved survival[29]. However, these findings could be influenced by factors like ALS-associated weight loss, hypermetabolism, and the survival benefits of a high-calorie diet[30,31]. Additionally, the cholesterol hydroxylase gene CYP27A1 has been genetically linked to ALS[32,33].

Motivated by a transcriptional signature of cholesterol overload in several ALS mouse models and ALS patient spinal cord, we tested the therapeutic potential of CD in our fast-progressing model for C9orf72 ALS, expressing GFP-(GA)175 using Nestin-Cre ("GA-Nes")[34]. This mouse line shows widespread poly-GA aggregates throughout the CNS, regional neuron loss, and microglia infiltration, leading to progressive weakness and weight loss. While poly-GA is expressed throughout the CNS, neurodegeneration starts in the hippocampal CA2 region and reaches spinal cord motor neurons at the endstage[34]. GA-Nes mice reach the humane endpoint at 6-7 weeks of age under national animal welfare laws[34]. Importantly, CD treatment extends the lifespan in female but not male mice. Single-cell transcriptomics of female mice suggests that poly-GA expression using the Nestin-Cre driver affects almost all cell types in the brain. Interestingly, the oligodendrocytes express several marker genes identified in DOLs in Alzheimer models[14], as well as ApoD and Plin4, which are tightly linked to lipid and cholesterol biology. CD treatment partially reverts the DOL signature and reduces Plin4 expression at the mRNA and protein level arguing for a beneficial effect of targeting altered lipid metabolism in the most common form of familial ALS.

## Results

### CD extends lifespan and reduces Neurofilament levels in female GA-Nes mice

Comparing our published gene expression datasets from two poly-GA mouse models (GA-Nes with widespread expression throughout the CNS and GA-CFP with expression mostly restricted to spinal neurons), the rNLS8 mouse line expressing TDP-43ΔNLS, and an updated large dataset of human ALS tissue[34–37] revealed a common signature of cholesterol dysmetabolism. Bulk RNAseq data (Fig. 1a) showed a reduction of key enzymes in the cholesterol biosynthesis pathway (e.g., Hmgcs1, Hmgcr), accompanied by an induction of the reverse cholesterol export pathway (Abca1, Abcg1). In addition, enzymes for cholesterol esterification (Soat1) and factors critical for lipid droplet formation (Plin2, Plin4) were upregulated, alongside several lipoproteins (Apoe, Apod, Apoc1). We confirmed our findings in unpublished dataset from independent cohorts of GA-CFP mice and rNLS8 mice (Fig. 1a). While we found no C9orf72-specific changes comparing

C9orf72 ALS and sporadic ALS cases, the cholesterol dysregulation signature was significantly more pronounced in male ALS patients. These changes suggest that overexpression of aggregating poly-GA and TDP-43 in mice leads to cholesterol overload, mirroring findings in the ALS spinal cord.

To test whether cholesterol overload contributes to ALS pathogenesis, we treated our rapidly progressive GA-Nes mouse model with 2-hydroxypropyl-β-cyclodextrin (hereafter referred to as CD). We administered CD (s.c. 2 g/kg) from weaning on postnatal day 21 daily until the humane endpoint was reached by weight loss or muscle weakness according to our standardized scoring system. The dose is in line with previous studies, but we administered CD daily rather than once or twice a week due to the fast progression in this mouse model[24,25]. Vehicle-treated male and female mice reached the endpoint at a median age of 41 and 43 days, respectively, in line with our initial characterization of this line (Fig. 1b, c)[34]. In contrast, half of the female GA-Nes mice receiving CD survived beyond 52 days (Fig. 1c). Possibly due to faster progression or due to pro-myelinating effects of testosterone[38] or due to differential lipid metabolism between genders, CD was much less beneficial in males, aligning with prior findings in female NPC mice[39]. These beneficial effects in female GA-Nes mice are reflected in a slower decline of body weight (Fig. S1).

To investigate the mechanism of CD-mediated lifespan extension, we used a second cohort of female GA-Nes treated from day 21 onwards and selected a fixed time point (postnatal day 40) for all further experiments to allow direct comparison in this rapidly progressing mouse model. Male and female transgenic mice showed strong upregulation of serum neurofilament light chain levels (Fig. 1d, e). This phenotype was significantly reduced by approximately 50% upon CD treatment in female GA-Nes mice (Fig. 1e), suggesting that CD attenuates neurodegeneration. In contrast, no such reduction was observed in CD-treated male mice (Fig. 1d), which aligns with the lack of survival benefits. Thus, we conducted all further experiments performed on GA-Nes mice solely on females.

CD treatment had no significant effect on the widespread poly-GA expression in GA-Nes mice (Fig. 1f, g). In contrast to previous immunohistochemistry characterization of poly-GA aggregates in paraffinized tissue, which was predominantly neuronal[34], GFP-(GA)175 expression was detected in most oligodendrocytes and astrocytes, but only a small subset of microglia in the cryosections imaged using native GFP fluorescence (Figs. 1f and S2), which is consistent with broad excision of the floxed stop cassette in the transgene due to widespread Nestin-Cre expression.

Taken together, GA-Nes mice show signs of cholesterol overload, and in female mice, life span is extended by daily treatment with the cholesterol-sequestering drug CD.

### CD mitigates poly-GA induced expression changes in oligodendrocytes

Since CD is known to have pleiotropic effects not only on cholesterol but also on inflammation, we conducted single-cell transcriptomics in the hippocampus of female mice at postnatal day 40 to investigate the mode of action. To facilitate comparison with emerging datasets from human ALS and FTD tissues, we chose to analyze single nuclei from frozen tissue. We prioritized the hippocampus for snRNA-seq, because neurodegeneration occurs first in the CA2 region and remains most pronounced there at the government-mandated humane endpoint[34]. Using flow cytometry, we isolated 10,000 DAPI-labeled nuclei per animal, 50% of which were NeuN-positive neurons and 50% were NeuN-negative glia (Fig. S3). The nuclei were processed on the 10X platform (Chromium Single Cell 3' Chip G v3.1). Metrics such as nuclei per sample, reads per nucleus, and reads per gene were consistent across all animals (Fig. S3a). After quality control, our dataset includes 90,772 nuclei from 8 animals in the wildtype condition and 66,263 nuclei from 7 animals in the transgenic condition (Supplementary Data 1).

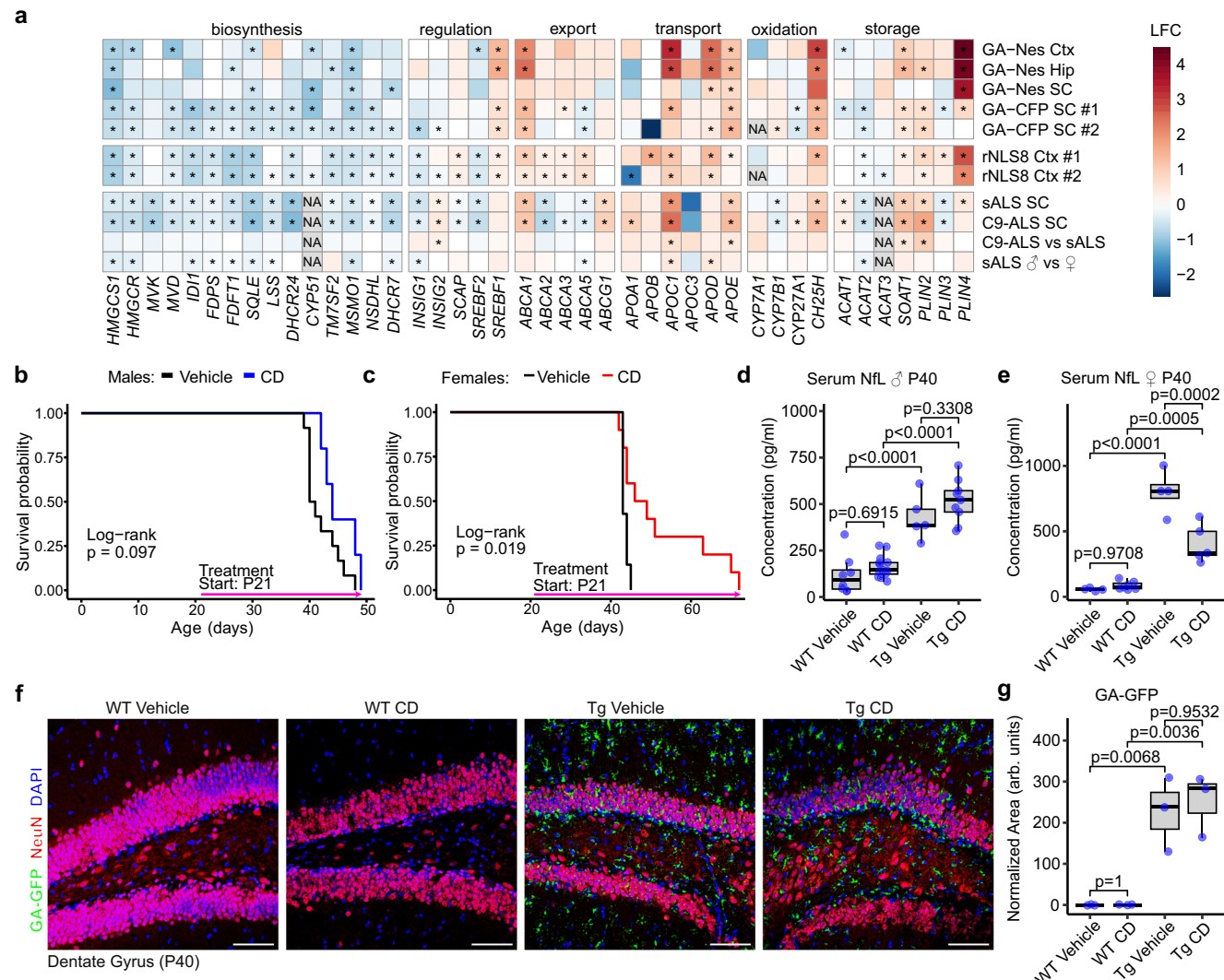

**Fig. 1 | Cyclodextrin extends lifespan and reduces Neurofilament levels in female GA-Nes mice. a** Gene expression changes in the cholesterol pathway from historical bulk RNAseq dataset in GA-Nes (endstage hippocampus, cortex and spinal cord, 5 control vs 5 GA-Nes mice), GA-CFP (thoracic spinal cord, cohort #1 with 4 wildtype vs. 3 GA-CFP mice 32 weeks of age, cohort #2 with 7 wildtype vs. 6 GA-CFP mice 42 weeks of age) and rNLS8 (hippocampus, after 3 weeks of transgene induction, cohort #1 with 17 wildtype vs. 10 rNLS8 mice, cohort #2 with 11 wildtype and 10 rNLS8 mice) ALS mouse models and patient spinal cord[34,37,51]. Log₂ fold changes compared to controls are shown in the heat map. Adjusted $p$ values from the original analysis are shown because different genotype and species prevent a combined re-analysis with common RNAseq pipelines. Asterisks indicate significant changes. Upregulation of export pathway genes and downregulation of synthesis genes suggest cholesterol overload. **b** Survival analysis by Kaplan–Meier curve shows that CD administered at 2 g/kg q.d. does not affect the survival of transgenic male mice significantly ($p = 0.097$ from log-rank test for

survival, $n = 12$ vehicle vs 5 CD). **c** Survival analysis by Kaplan–Meier curve shows that CD administered at 2 g/kg q.d. significantly improves the survival of transgenic female mice ($p = 0.019$ from log-rank test for survival, $n = 7$ vehicle vs 10 CD). **d, e** Serum NfL from independent cohorts of mice sacrificed at postnatal day 40 (P40) shows significant modulations (female $n$ from independent biological replicates: WT Vehicle = 4, WT CD = 7, Tg Vehicle = 4, Tg CD = 5; male $n$ of independent biological replicates: WT Vehicle = 8, WT CD = 15, Tg Vehicle = 5, Tg CD = 9). Post-hoc analysis by Tukey's HSD shows a dramatic increase in transgenic mice, which is partially rescued by CD treatment in female but not male mice. CD administration does not modulate wild-type NfL levels. **f, g** Fluorescence microscopy of endogenous GA-GFP in frozen tissue confirms its neuronal and glial expression in transgenic mice and the lack of expression in wild-type mice, while also highlighting no significant reduction upon CD treatment (ANOVA, post hoc by Tukey's HSD from three independent biological replicates in each group). All scale bars = 75 μm.

Clustering of all cells revealed two states of microglia (proliferating vs. non-dividing), four states of oligodendrocytes (OPC, newly formed oligodendrocytes and myelinating Oligo_mat1 and Oligo_mat2) and 18 subtypes of neurons (Fig. 2a). The expression of the five best marker genes per cluster is shown in Fig. S4, with the absolute number of cells per cluster detailed in Supplementary Data 1.

Although the comparison of cell numbers is limited by the NeuN-based enrichment strategy, it is clear that poly-GA expression strongly increased the number of microglia, including a proliferating subcluster (Figs. 2a and S4, S5, Microglia_div, top markers: Rad51b, Ly86, Atad2, Diaph3, Lrmda). This resulted in an undersampling of other glia, with

an apparent lower abundance of astrocytes and oligodendrocytes. Importantly, our histological quantification of cell density in Fig. S6 shows a ~5-fold increase in Iba1⁺ microglia, which was not significantly affected by CD treatment. While oligodendrocyte numbers were not unaffected by poly-GA expression or CD treatment, the number of S100β-positive astrocytes was even slightly increased in GA-Nes mice, with a trend towards reduction upon CD treatment. Rare clusters of macrophages and other immune cells (B cells, T cells, and a few NK cells) also appeared highly enriched in poly-GA expressing mice in our snRNAseq analysis (Fig. S6). In addition, the relative number of nuclei in the prominent neuroblast population at postnatal day 40 was

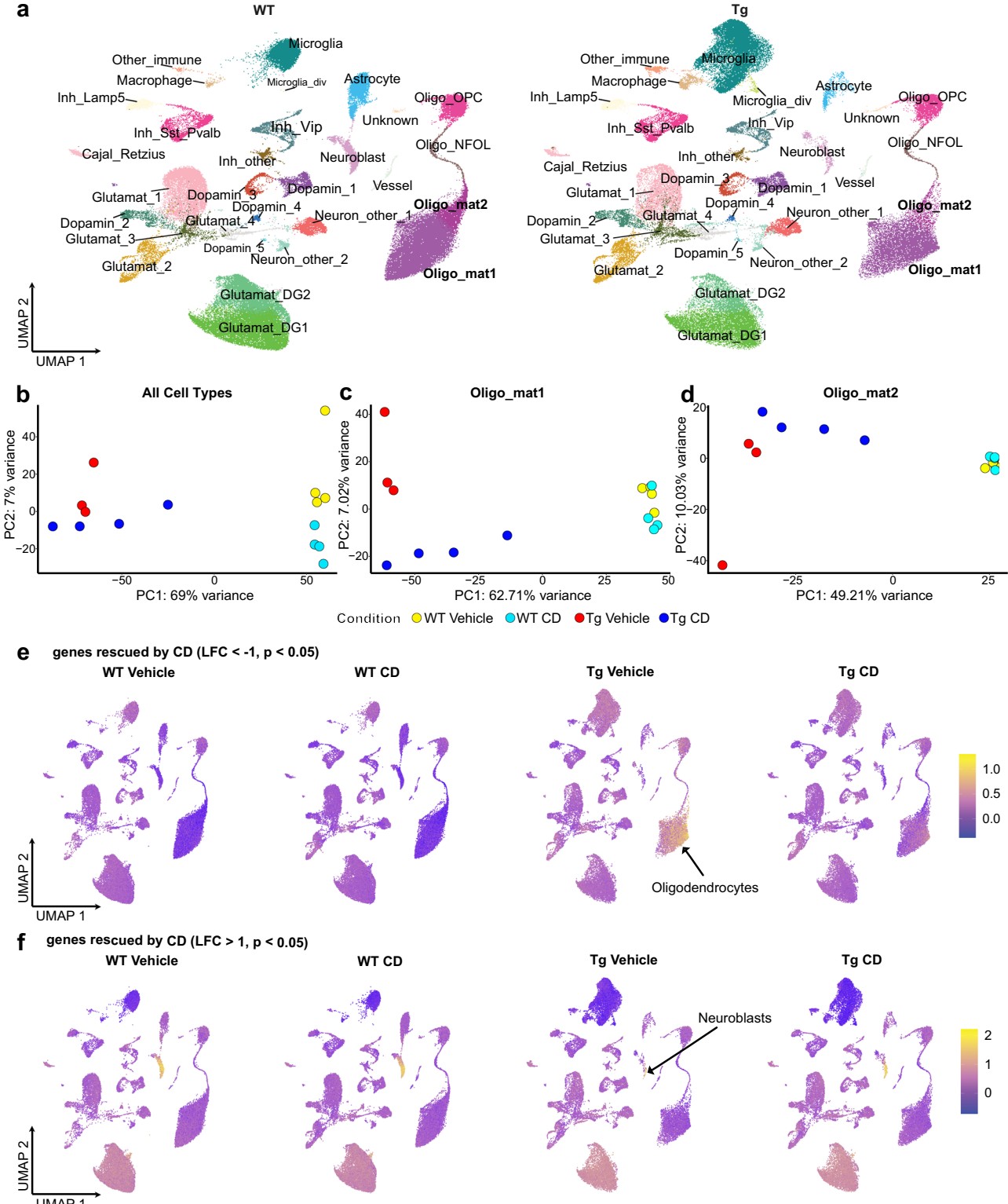

**Fig. 2 | snRNAseq highlights oligodendrocytes as target of CD therapy by downregulation of genes. a** snRNAseq dimensional reduction plot (UMAP) of WT (90,772 cells from 8 animals) and Tg (66,263 cells from 7 animals) highlights dramatic changes in glial cells. **b** PCA of all expressed genes by combining all clusters clearly separates the genotypes and partially resolves effects by CD treatment. **c**, **d** PCA of all expressed genes in Oligo_mat1 and Oligo_mat2 clusters, respectively, highlights the rescue effect of CD treatment in the clusters of mature oligodendrocytes. **e** FeaturePlot of genes combined with AddModuleScore that are downregulated (LFC < −1, adjusted $p < 0.05$) by CD treatment shows that the target of such therapy is primarily oligodendrocytes. **f** FeaturePlot of genes with color-coded AddModuleScore that are upregulated (LFC > 1, adjusted $p < 0.05$) by CD treatment shows that the target of such therapy is primarily neuroblasts.

reduced in GA-Nes mice (Fig. S5). Quantification of neuroblasts with DCX immunofluorescence confirmed a ~80% loss of neuroblasts in GA-Nes mice, which was not rescued by CD treatment (Fig. S6). In line with previous findings, we also detected pronounced neuron loss in the hippocampal CA2 region, which was, however, not significantly rescued by CD treatment (Fig. S6).

Principal component analysis (PCA) analysis of all cell types distinctly separated wild-type and transgenic mice (Fig. 2b and Supplementary Data 2) in the first component, as well as treated and untreated mice in the first and second components. The clustering of CD and vehicle-treated GA-Nes mice was most evident for the mature oligodendrocyte clusters (Fig. 2c, d) and less prominent for other cell types (Fig. S7). Interestingly, transgenic and wildtype mice are clearly separated by the PCA in all subsets, except in the two dopaminergic neuron types (clusters 4 and 5).

Correlation analysis on pseudo-bulk data from each subcluster revealed that gene expression changes induced by poly-GA were generally positively correlated across all cell types, especially among the neuronal subtypes (Fig. S8). CD treatment partially reversed these effects, as indicated by the negative correlation, with the most substantial impact on the Oligo_mat1 and Oligo_mat2 populations and neuroblasts. Moreover, filtering all poly-GA upregulated genes that were significantly rescued by CD treatment (e.g., Neat1, ApoD, Plin4, Rhoj) and plotting the UMAP highlights oligodendrocytes as the main target of CD (Fig. 2e). A similar analysis of all poly-GA downregulated genes that are rescued by CD (e.g., Cntnap5a/b, Zbtb20, Plxna2, Sema5a) showed a strong enrichment in neuroblasts average expression (Fig. 2f). Importantly, CD treatment also significantly reduced the cholesterol dysfunction in oligodendrocytes based on cell type-specific analysis of that pathway (Fig. S9). Interestingly, cholesterol export and storage pathways were mostly induced in oligodendrocytes of GA-Nes mice and less consistently in microglia and astrocytes.

Taken together, poly-GA expression induces a pronounced microgliosis in GA-Nes mice, but also affects all other cell types. CD treatment most strongly affects gene expression in oligodendrocytes and neuroblasts in transgenic mice, but has little effect in controls (Fig. S7). Since the high density of neuroblast in young GA-Nes mice is unlikely to reflect ALS/FTD pathology in aged patients, we focused our further analysis on oligodendrocytes.

### CD ameliorates DOL formation in GA-Nes mice

Disease-associated oligodendrocytes (DOLs) with a common gene expression signature have recently been identified in mouse models of Alzheimer's disease, demyelination, and aging[14]. Serpina3n, C4b, Il33, and other DOL marker genes were significantly upregulated in the oligodendrocyte clusters of GA-Nes mice (Figs. 3a and S10). Many of these genes were also significantly induced in previous bulk sequencing data[34] from hippocampus, neocortex and less severely at in the spinal cord of GA-Nes mice at the humane endpoint (Fig. 3a). Volcano plots highlight additional genes (e.g., Kcnma1, Apod, Plin4) that are strongly induced in GA-Nes mice and significantly rescued by CD treatment (Fig. 3b). Immunofluorescence staining and immunoblotting with Serpina3n further confirmed a strong induction of DOLs in GA-Nes mice, which was significantly rescued by CD treatment (Fig. 3c−e). We confirmed the rescue of IL33 expression using an immunoassay (Fig. 3f). Enhanced expression of the chemokine CXCL10/IP10 in GA-Nes mice was also reduced by CD treatment (Fig. S11), while other cytokines/chemokines were unaffected by CD (Fig. S11 and S12).

### CD mitigates demyelination-related pathology in GA-Nes mice

To elucidate the mode of action for CD therapy in GA-Nes mice, we analyzed gene ontology terms associated with expression changes rescued by CD treatment across different cell types. Using a cutoff of

|LFC| > 0.5, revealed enrichment of the myelin pathway in oligodendrocytes (Oligo_mat1), while axon and synapse formation was stimulated in neuroblasts (Fig. 4a). The major protein components of myelin were transcriptionally downregulated in the Oligo_mat1 and 2 populations and these changes were partially rescued by CD, particularly in the Oligo_mat1 subpopulation. (Fig. 4b).

Microglia adopt different states in response to stimuli such as Aβ aggregates and demyelination, often termed disease-associated microglia (DAM)[40,41]. DAM microglia, however, differ in their molecular make-up in response to different stimuli and can thus be segregated into "myelin-DAM" and "amyloid-DAM"[40]. Notably, microglia in GA-Nes mice exhibited a significant upregulation of the myelin-DAM signature. Following CD treatment, this signature shifted towards amyloid-DAM, indicating a reduction in myelin debris (Fig. 4c).

To assess myelination directly, we analyzed female GA-Nes mice at P40 by electron microscopy (Fig. 4d, e). The corpus callosum was thinner, and the density of myelinated axons was greatly reduced compared to wild-type littermates. We observed some swollen axons with organelle accumulation (Fig. S13a), but large lipid droplets were absent in microglia, neurons, and oligodendrocytes. Importantly, the density of myelinated axons was significantly increased in CD-treated GA-Nes mice, suggesting that CD supports oligodendrocyte function by restoring cholesterol homeostasis (Fig. 4d, e). To distinguish between poly-GA-induced demyelination and developmentally impaired myelination, we quantified the density of myelinated axons at postnatal day 21, the time we had started CD treatment in our previous experiments. Myelination in GA-Nes mice was not significantly reduced at this young age compared to littermate controls, although there was a small trend (Fig. S13b, c), which is consistent with very modest changes at the transcriptome level at this age[34]. This suggests that GA-Nes mice undergo active demyelination between P21 and P40, which is significantly ameliorated by CD treatment. However, since the density of myelinated axons increases significantly between P21 and P40 in wildtype mice (and to a lesser extent in transgenic mice), we cannot completely exclude the possibility of a poly-GA-dependent inhibition of developmental myelination during this period.

Given the high cholesterol content of myelin, we analyzed cholesterol and cholesterol esters by mass spectrometry. Total levels of unmodified cholesterol were decreased in GA-Nes animals compared to controls, suggesting a general loss of myelin, possibly already during development. In contrast, cholesteryl esters (especially with monounsaturated 18-carbon fatty acid, abbreviated as CE 18:1) were greatly increased in GA-Nes animals, with a significant rescue in CD-treated mice (Fig. 4f, g). This could reflect reduced accumulation of myelin debris, alternative metabolic processing, or excretion from the CNS.

In the absence of robust human-derived neuron/oligodendrocyte co-culture models, we overexpressed (GA)$_{149}$-GFP in iPSC-derived human neurons to investigate a potential direct effect of poly-GA on cholesterol metabolism. (GA)$_{149}$-GFP formed aggregates (Fig. S13d) similar to previous experiments in primary rat neurons[42]. Poly-GA expression resulted in a minimal but significant reduction of several cholesterol biosynthetic enzymes compared to the GFP control, without affecting lipid export or storage pathways (2.5 mg/mL from day 10 to day 19) (Fig. S13e). Interestingly, CD treatment significantly increased cholesterol biosynthetic pathways in both (GA)$_{149}$-GFP and GFP-expressing cells, suggesting that CD depletes the neuronal cholesterol pool, leading to compensatory upregulation of biosynthetic pathways. These findings are consistent with the minimal alterations in cholesterol biosynthesis, export, and storage pathways observed in neuronal populations of GA-Nes mice (see above, Fig. S9). This argues against poly-GA-induced accumulation of excess cholesterol in neurons, in contrast to the case of oligodendrocytes.

Taken together, GA-Nes mice show pronounced axon and myelin damage, evidenced by the accumulation of cholesteryl esters. CD

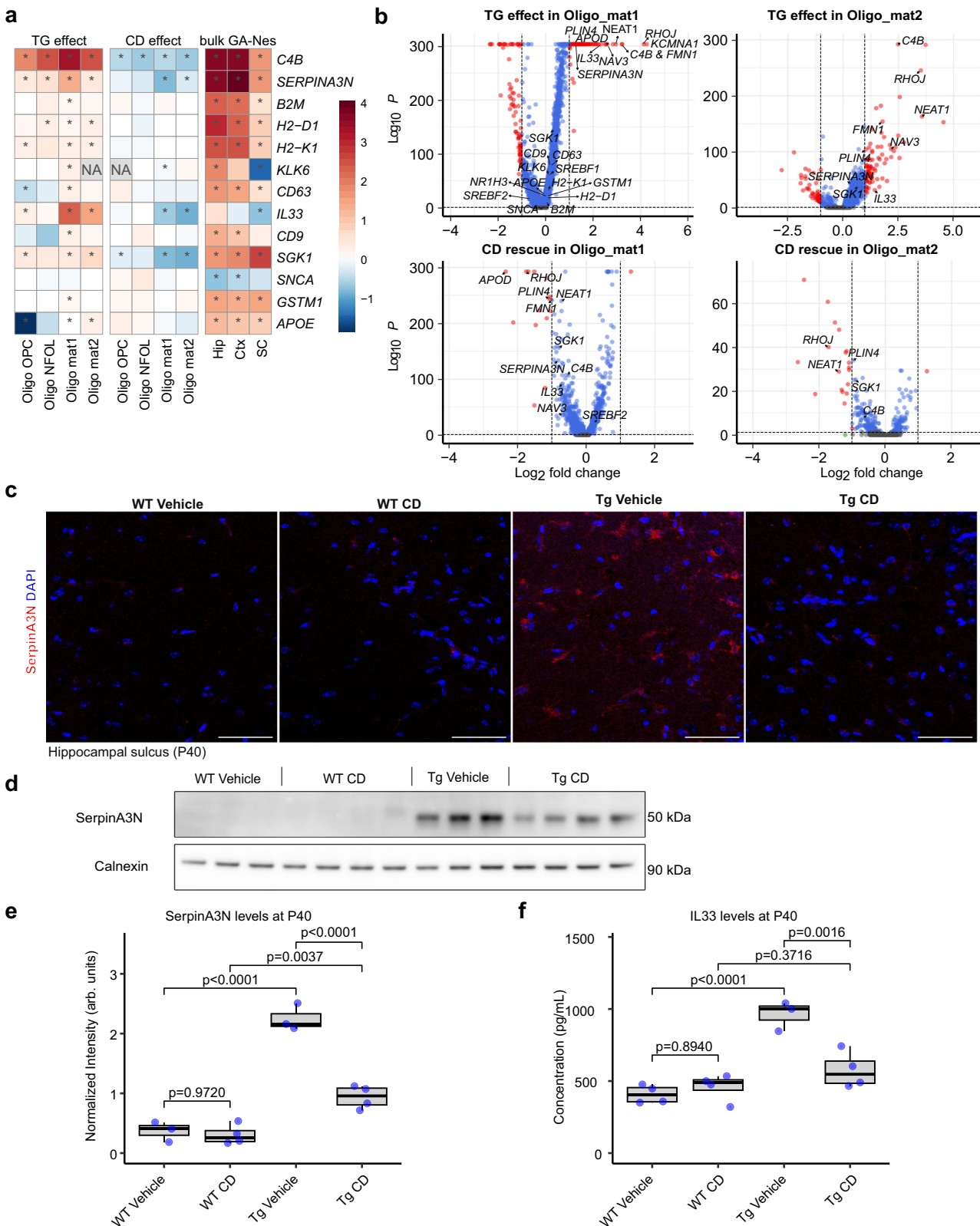

**c** Hippocampal sulcus (P40)

**d** SerpinA3N 50 kDa / Calnexin 90 kDa

**e** SerpinA3N levels at P40

**f** IL33 levels at P40

treatment mitigates these detrimental effects, underscoring its therapeutic potential.

## Plin4 upregulation in oligodendrocytes of GA-Nes mice is rescued by CD

Among the genes rescued by CD treatment, Perilipin (Plin4) caught our attention, because it associates with lipid droplets, containing cholesterol esters, and its expression was minimal in wild-type mice. Plin4 was strongly induced in mature oligodendrocytes of female transgenic mice, which was partially reversed by CD (Fig. 5a, b). Double immunofluorescence identified Plin4 primarily in carbonic anhydrase II (CA2)-positive oligodendrocytes of transgenic mice, with negligible expression in Iba1-positive microglia and GFAP-positive astrocytes of transgenic mice (Figs. 5c and S14). Plin4 staining was most pronounced

**Fig. 3 | poly-GA transgene induces a DOL signature which is rescued by CD treatment. a** Heatmap of previously published DOL genes[14] across oligodendrocyte clusters comparing transgenic effect ($n$ = 3 Tg Vehicle vs $n$ = 4 WT Vehicle) and CD effect ($n$ = 4 Tg CD vs $n$ = 3 Tg Vehicle) as well as bulk RNAseq from the endstage GA-Nes mice (data from[34], 5 control vs 5 GA-Nes mice) demonstrates induction of DOL genes in the diseased state. CD treatment showcases an appreciable and significant rescue of many of these genes. Asterisks indicate significant changes ($p$ < 0.05). **b** Volcano plots of the two mature oligodendrocyte clusters demonstrate some overlap with previously published DOL genes as well as other genes such as ApoD, where they get rescued by CD treatment. **c** SerpinA3N immunofluorescence demonstrates that increased expression of this key DOL marker in transgenic GA-Nes

mice is curtailed by CD treatment, confirming the snRNAseq results. Scale bars = 50 μm. Images are representative of two independent experiments. **d, e** Western blot of hindbrains ($n$ from independent biological replicates: WT vehicle = 3, WT CD = 4, Tg Vehicle = 3, Tg CD = 4) and its quantification confirms the significant induction of Serpin A3N upon transgene expression and its significant rescue upon CD treatment (ANOVA, post hoc by Tukey's HSD). **f** IL33, another key DOL marker, measured by MSD V-Plex cytokine panel from hindbrain lysates shows significant modulation of this marker. Post-hoc analysis with Tukey's HSD highlights upregulation of IL33 in the transgenic condition, which is fully rescued upon CD treatment. The treatment does not modulate IL33 levels in wild-type mice. n from independent biological replicates: WT Vehicle = 4, WT CD = 4, Tg Vehicle = 3, Tg CD = 4.

in the cytoplasm of the cell body. Quantification confirmed that ~40% of CA2-positive oligodendrocytes contained Plin4 in transgenic animals, which is reduced to less than 10% in CD-treated animals matching WT expression (Fig. 5d). Taken together, poly-GA in transgenic mice induces the expression of the lipid droplet protein Plin4 predominantly in oligodendrocytes, which is partially rescued by CD treatment and may reflect cholesterol overload in the untreated GA-Nes mice.

### Neuronal poly-GA and TDP-43ΔNLS induce Plin4 expression and DOL marker expression

To exclude aberrant effects of Nestin-Cre driven poly-GA expression in oligodendrocytes, we used Camk2a-Cre to express the poly-GA transgene exclusively in excitatory neurons[43]. These "GA-Camk2a" mice developed similar phenotypes as the GA-Nes mice, albeit at a slower pace, reaching the humane endpoint at approximately 30 weeks (Zhou et al., manuscript in preparation). In addition, we investigated the rNLS8 mouse model expressing TDP-43ΔNLS driven by a neurofilament heavy chain promoter, resulting in almost exclusive neuronal expression and some very sparse expression in oligodendrocytes[36]. Bulk transcriptomics analysis confirms induction of the DOL signature and cholesterol dysmetabolism, including Plin4 induction, in 30-week-old GA-Camk2a mice and 3 weeks after transgene induction in rNLS8 mice (Fig. 6a). Immunofluorescence confirmed strong induction of Serpina3n expression in the hilus of the dentate gyrus of rNLS8 mice (Fig. 6b, c). Similarly, Serpina3n expression was significantly induced in GA-Camk2a mice (Fig. 6d).

 Finally, we confirmed the induction of Plin4 in oligodendrocytes of GA-Camk2a mice using immunofluorescence in white and gray matter of the hippocampus (Fig. 6f). In GA-Camk2a white matter, Plin4 expression was in the soma of CA2-positive oligodendrocytes (white arrow, Fig. 6f) and colocalized with the axon tracts (yellow arrow). In addition, we noticed Plin4 expression in the neuropil in white and gray matter (magenta arrow in Fig. 6f).

 Taken together, Plin4 expression, DOL induction, and cholesterol dysmetabolism are conserved in a second poly-GA mouse model and a TDP-43 model with predominantly neuronal expression of aggregates.

## Discussion

While ALS is not a primary demyelinating disease, significant neurodegeneration leads to myelin loss, most prominently in the corticospinal tract. Excess cholesterol from myelin loss must be stored as cholesteryl esters in the form of lipid droplets or removed from the CNS, mostly as 24S-hydroxy cholesterol[26,44]. We explored the therapeutic potential of the cholesterol-sequestrating drug CD in a C9orf72 mouse model with widespread poly-GA expression characterized by impaired cholesterol clearance and myelin loss. CD extends lifespan in female GA-Nes mice and reduces neurodegeneration markers, improves myelination, and mitigates expression of the DOL signature, including levels of the lipid droplet component Plin4. Cholesterol dysmetabolism and formation of DOLs are confirmed in a second poly-GA mouse model and a TDP-43 model with pure neuronal poly-GA expression.

### Cholesterol dysmetabolism and microglia response in ALS

Various poly-GA and TDP-43 mouse models, along with sporadic and C9orf72 ALS patient spinal cords, show upregulation of the cholesterol export pathway (ApoE and Abca1) and a decrease in new cholesterol synthesis. This pattern resembles an LXR-driven response to an overload of toxic free cholesterol, although total cholesterol levels were even reduced in GA-Nes, which likely reflects the compromised myelination. The upregulation of C16 and C18 cholesteryl esters in GA-Nes mice strongly indicates enhanced storage of excess cholesterol in lipid droplets and aligns with findings from SOD1 mice and ALS patient tissue[45–47].

 Cholesterol overload with subsequent LXR-response has been described in response to injury and demyelination[23,48]. For example, induction of acute demyelination with lysolecithin in wild-type mice leads to cholesterol accumulation in microglia, resulting in a "foam cell" morphology due to uptake of cholesterol-rich myelin debris[23]. As in the lysolecithin model, inefficient clearance of cholesterol may limit remyelination and thus enhance disease progression in ALS patients. Exhausting the phagocytic capacity of microglia with myelin debris has been shown to alter the microglia response towards other targets, such as Aβ[40]. Interestingly, the "myelin-DAM" (disease-associated microglia) signature described in demyelination models is clearly enriched in poly-GA and TDP-43 mouse models, while CD treatment reduces cholesteryl ester levels and shifts the microglia phenotype to the "amyloid-DAM" originally identified in Alzheimer's mouse model[40,41]. Unfortunately, snRNAseq data is less suited for subclustering of microglia than single-cell RNAseq data due to poor sensitivity for many key marker genes[49]. Intracellular poly-GA and TDP-43 aggregates in ALS activate microglia, which may promote clearance or inhibit cell-to-cell transmission[34,37,50,51].

 Excess cholesterol forms crystals that activate the inflammasome. Hydroxylated cholesterol derivates, in particular 24S-hydroxy cholesterol, can be excreted from the CNS, but in case of clearance failure, can also be toxic to oligodendrocytes at high concentrations[44,52]. Moreover, auto-oxidized cholesterol derivatives that harm motoneuron health have been identified in SOD1 models and ALS tissue[53]. Although cholesteryl esters are generally inert, the acyl group donors needed for their synthesis, like phosphatidylcholine, can convert into toxic byproducts, such as lysolecithin[46]. CD may inhibit all these pathways by sequestration of cholesterol, resulting in reduced DOL formation and altered microglial phenotype, suggesting therapeutic potential in modulating cholesterol-induced pathology. Interestingly, CD leads to compensatory induction of cholesterol biosynthesis and cholesterol uptake in control iPSC-derived neurons confirming the cholesterol clearance function of CD.

### Lipid droplets and Plin4 in ALS

PLIN4 is a poorly characterized member of the perilipin family. It preferentially binds to cholesteryl esters and is expressed mainly in adipose tissue and muscle under physiological conditions[54]. Interestingly, pathogenic mutations in Plin4 can trigger its aggregation in

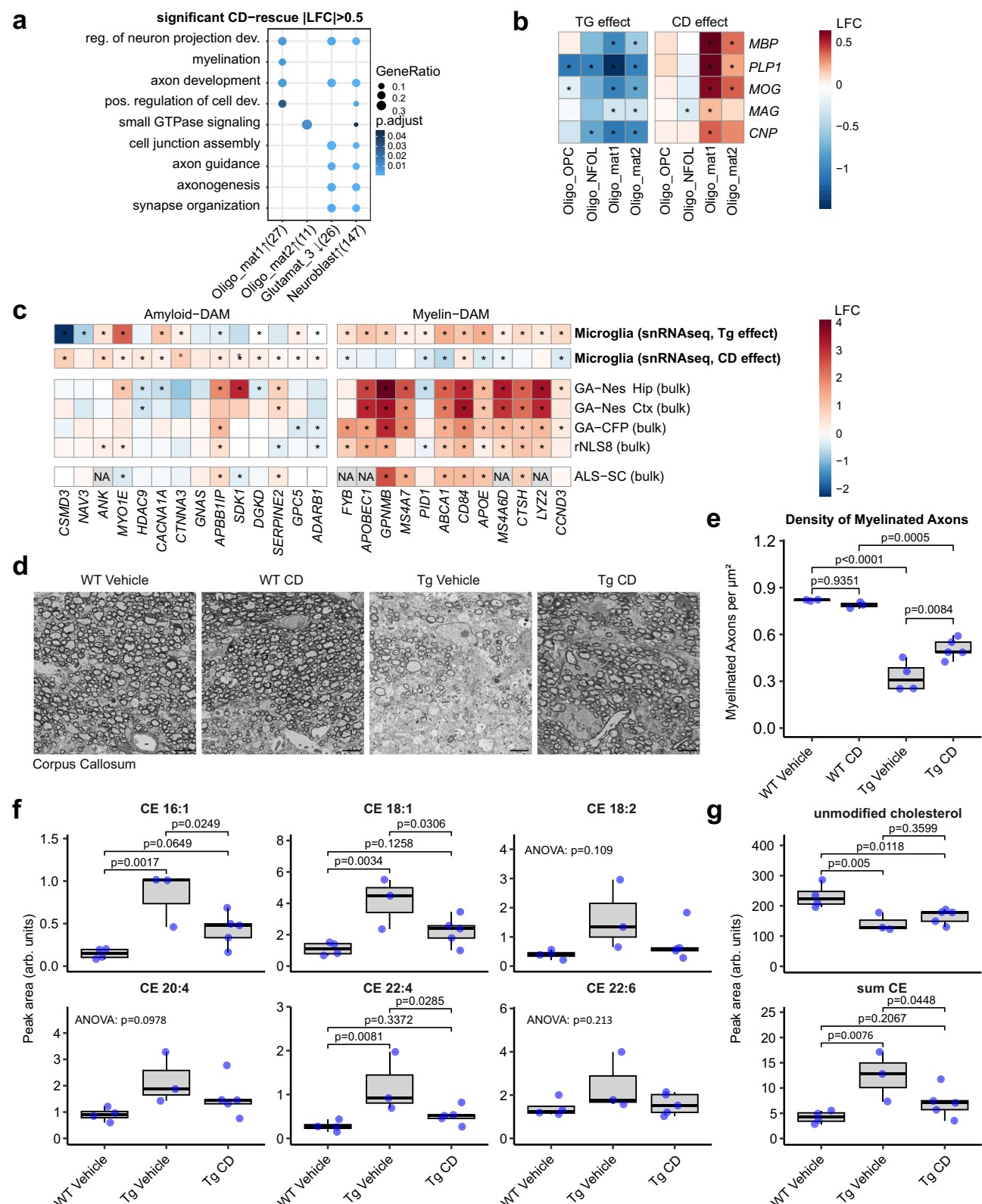

muscle, resulting in a myopathy[55]. Perilipins, including PLIN4, coat lipid droplets and regulate their decomposition through lipases and lysosomes[56]. Moreover, PLIN4 droplets have been observed in neurons and astrocytes in Parkinson's disease mouse model induced by MPP+ [57]. The knockdown of Plin4 in SH-SY5Y cells mitigates MPP+ toxicity by resorting mitophagy, suggesting lipid droplets overwhelm the autophagy machinery.

In the context of ALS, Plin4 upregulation has only been reported in SOD1[G93A] mice, where it starts before symptom onset and continues to increase to the late stage[58]. However, Plin4 is predominantly expressed in neurons in the SOD1[G93A] mice, while it is primarily expressed in oligodendrocytes in GA-Nes and GA-Camk2a mice. In addition, PLIN4 expression in oligodendrocytes has been reported in the experimental autoimmune encephalomyelitis mouse model[18],

**Fig. 4 | CD mitigates demyelination-related pathology in GA-Nes mice. a** Gene ontology of genes rescued by CD with |LFC| > 0.5 shows that upregulated genes in Oligo_mat1 and neuroblast cluster are contributing to myelination and axonal development. The number of genes per cell type is indicated in parenthesis. **b** Heatmap of gene expression of major myelin protein components demonstrates downregulation of myelination in all oligodendrocyte clusters as the result of transgene expression (left block), an effect which is significantly rescued with CD treatment (right block). Asterisks indicate significant changes (adjusted $p < 0.05$). **c** A myelin DAM signature is detected in various models of ALS as well as patient spinal cord. Microglia cluster also strongly shows this signature, while CD administration partially alleviates this signature, shifting it more towards the amyloid DAM signature. Asterisks indicate significant changes (adjusted $p < 0.05$). **d, e** SEM overview and quantification of corpus callosum shows dramatic loss of myelinated axons upon transgene expression, which is partially but significantly rescued by CD treatment (ANOVA, post hoc by Tukey's HSD). $n$ from independent biological replicates: WT Vehicle = 3, WT CD = 3, Tg Vehicle = 4, Tg CD = 5. Scale bars = 3 μm. **f** Measurements of cholesteryl ester species by mass spectrometry demonstrate significant upregulation in transgenic condition and CD-mediated rescue of CE 16:1, CE 18:1, and CE 22:4. CE 18:2, CE 20:4, and CE 22:6 were not significantly modulated (ANOVA with Fisher's LSD post-hoc). $n$ from independent biological replicates: WT Vehicle = 4, Tg Vehicle = 3, Tg CD = 5. **g** Total unmodified cholesterol and cholesteryl esters (from peak area sum) confirm downregulation of cholesterol in transgenic condition, which is not affected by CD. Cholesteryl esters are upregulated in GA-Nes mice, which is rescued by CD (ANOVA with Fisher's LSD post-hoc. $n$ from independent biological replicates: WT Vehicle = 4, Tg Vehicle = 3, Tg CD = 5).

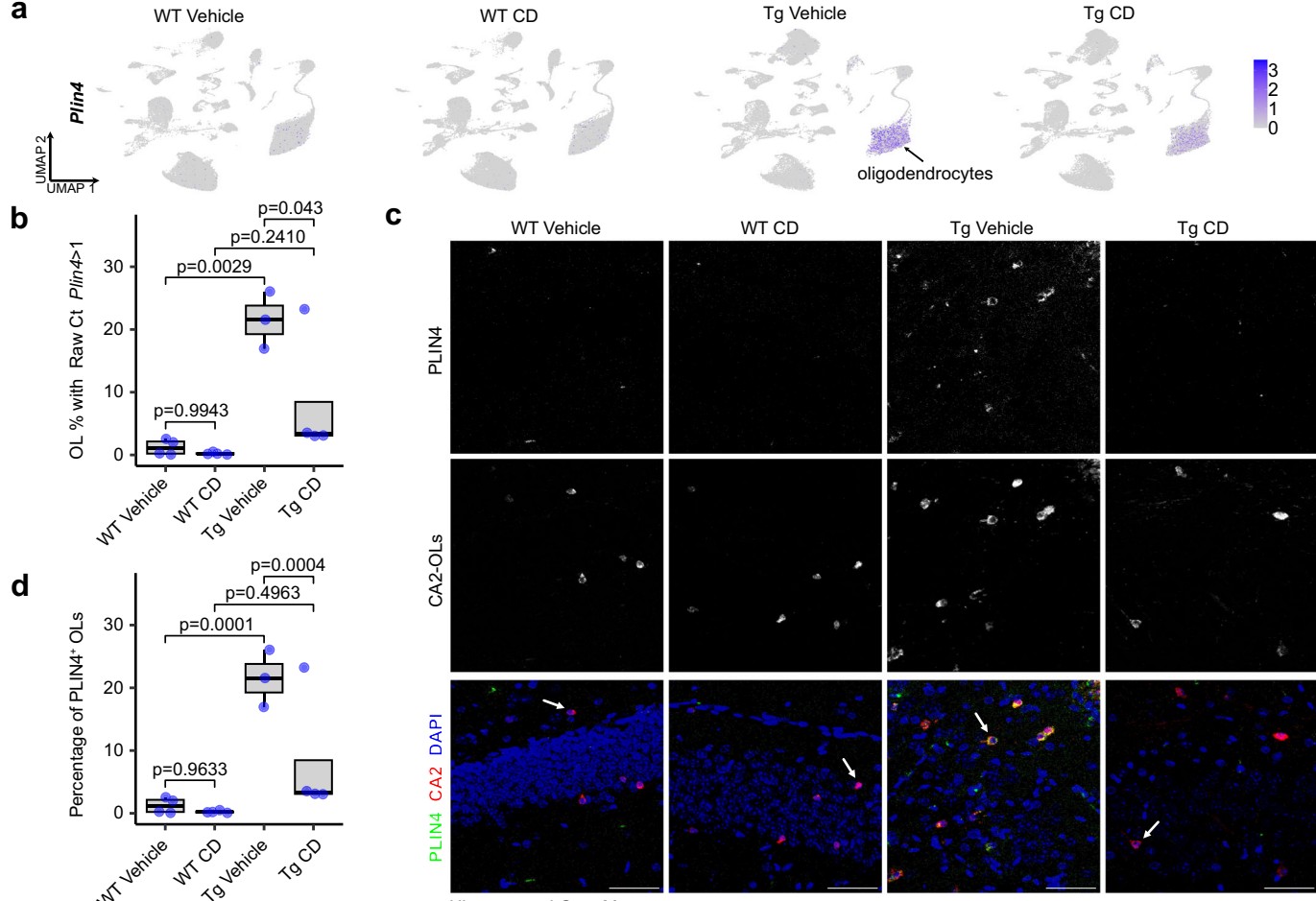

**Fig. 5 | Plin4 production is upregulated in oligodendrocytes, and rescued by CD treatment. a** FeaturePlot of PLIN4 shows upregulation of its expression in GA-Nes mice and rescue by CD treatment. Major glia types are clustered as indicated. **b** Percentage of mature oligodendrocytes expressing more than one copy of PLIN4 (raw count) is dramatically increased with the expression of the GA transgene, and partially rescued by CD treatment. Data is generated per each animal from the snRNAseq data (ANOVA, post hoc by Tukey's HSD; $n = 4$ WT Vehicle, $n$: WT CD, $n = 3$ Tg Vehicle, $n = 4$ Tg CD). **c** PLIN4 (green) immunofluorescence co-stained with DAPI and CA2 (red) confirm the transgenic upregulation and rescue by CD. White arrows indicate examples of CA2+ oligodendrocytes. Scale bars = 50 μm; Shown images are adjusted for brightness and contrast in the same way for each channel across all conditions. **d** Quantification of percentage of PLIN4+ oligodendrocytes in hippocampus highlights a dramatic increase of such oligodendrocytes with the expression of the transgene, and a near-complete rescue by CD treatment. $n = 3$ in each of the four conditions, from independent biological replicates (ANOVA, post hoc by Tukey's HSD). Quantification was performed exclusively on raw images.

mirroring the subcellular distribution pattern we observed. These findings, together with our data, support a potential role for Plin4 in primary and secondary demyelinating diseases.

Despite strong upregulation of Plin4, we and others have not observed large lipid droplets in oligodendrocytes under demyelinating conditions. To date, lipid droplets, in this case labeled by Plin1, have only been reported in oligodendrocytes expressing the AD risk allele ApoE4, which is mitigated by CD treatment in both cellular and animal models[15]. Although we did not observe cholesterol crystals or excessive lipid droplets in microglia or oligodendrocytes through (electron) microscopy, lipidomic analysis confirms a strong increase in cholesteryl esters in GA-Nes mice, which is normalized by CD treatment.

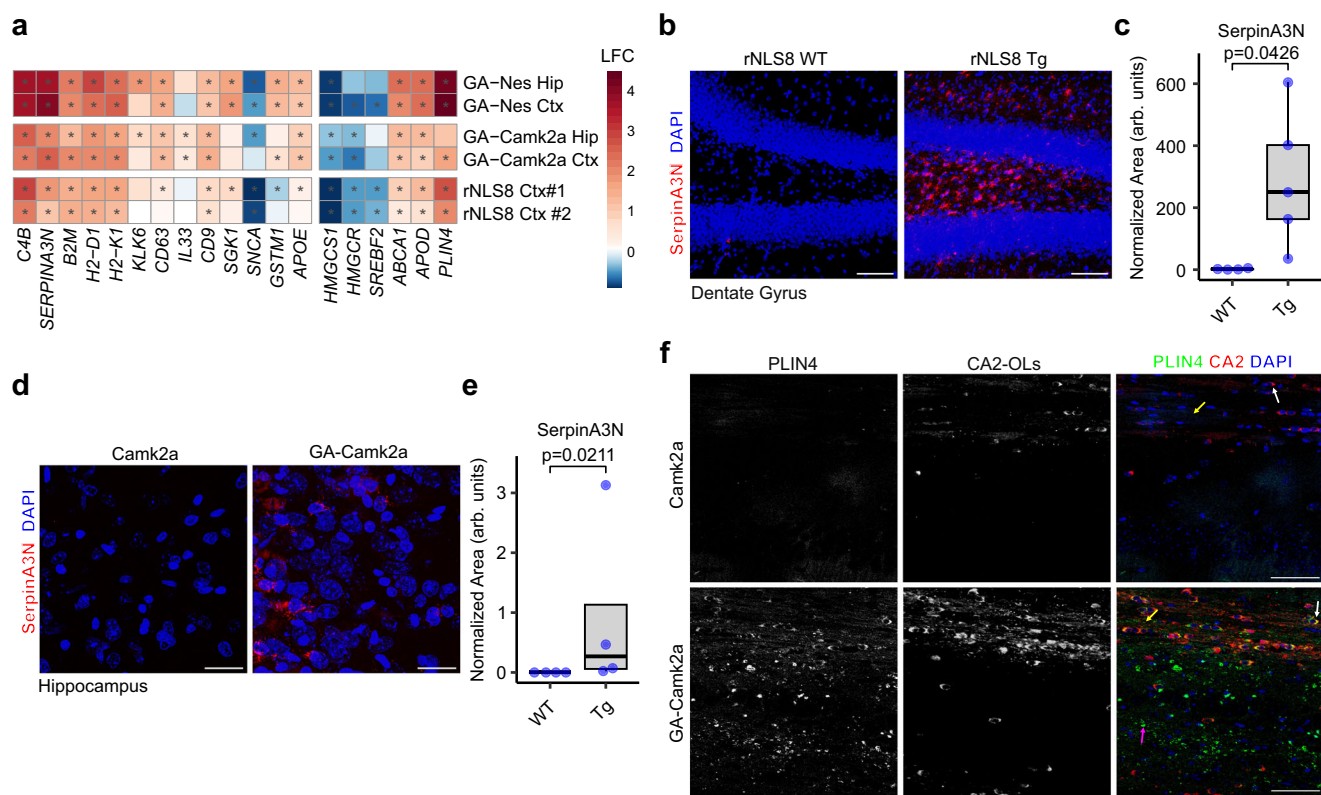

**Fig. 6 | DOLs are found in other ALS mouse models. a** Heatmap of DOL marker genes and essential cholesterol pathway genes from bulk RNAseq across three mouse models of ALS confirms induction of DOLs and the disturbance of cholesterol pathway. Asterisks indicate significant changes (adjusted $p < 0.05$). ($n$ as Fig. 1a for GA-Nes and rNLS8 mice. GA-Camk2a hippocampi from 5 control vs 6 transgenic mice. GA-Camk2a neocortex from 4 control vs 6 transgenic mice). **b, c** SerpinA3N staining in rNLS mice shows significant expression of Serpina3n mRNA in the transgenic condition. n: WT = 4, Tg = 5 from independent biological replicates. (Welch's $T$-test) **d, e** SerpinA3N staining in GA-Camk2a mice shows significant expression of Serpina3n mRNA in the transgenic condition. $n$: WT = 4, Tg = 4 from

independent biological replicates (Mann–Whitney $U$-Test). **f** PLIN4 (green) immunofluorescence co-stained with DAPI and oligodendrocyte marker CA2 (red) in Camk2a (control) and GA-Camk2a (transgenic) mice, respectively, confirms upregulation of PLIN4 in transgenic mice, both in oligodendrocytes as well as elsewhere. The shown images are adjusted for brightness and contrast in the same way for each channel in similar ways across both conditions. White arrow indicates an example of a CA2+ oligodendrocyte, yellow arrow shows an example of axon tracts, while magenta arrow highlights an example of PLIN4 in the neuropil. Images are representative of three animals. Scale bar = 50 μm.

## DOLs and Oligodendrocyte dysfunction in ALS

The fascinating discovery of a conserved signature of DOLs in mouse models of Alzheimer's disease and multiple sclerosis has sparked a new interest in oligodendrocyte biology in other neurodegenerative diseases and mouse models[14,18,19]. It is unclear whether DOLs promote neurodegeneration or reflect a protective response.

Emerging single-nucleus RNAseq data from ALS patients show that the expression of myelin-encoding genes is reduced in oligodendrocytes in the frontal cortex of C9orf72 FTD/ALS and the motor cortex of ALS cases[59,60], and myelination deficits have been reported[61,62]. In addition, oligodendrocyte deficits have been documented in several ALS mouse models, including SOD1 transgenic mice[13], FUS knockout mice[63], and TDP-43 knockout mice[64]. A very recent study reported reduced myelination and cholesterol biosynthesis upon oligodendrocyte-specific expression of mutant TDP-43 M337V in mice, but DOL formation was not investigated[17]. TDP-43 aggregates are also common in oligodendrocytes, particularly in C9orf72 ALS[4,5], and C9orf72 and SOD1 mutations also have direct effects on oligodendrocytes that may harm motor neurons by promoting lactate release[65]. Interestingly, while DPR aggregates have so far only been detected in neurons and ependymal cells, C9orf72 is also widely expressed in glial cells including oligodendrocytes[7,8,66,67].

The exact pathway of DOL induction is still unclear, but may involve inflammatory pathways such as STAT and NFκB in myelinating oligodendrocytes[14]. GA-Nes mice show a robust DOL signature in

snRNAseq data, with several established markers verified at the protein level. In our initial characterization of the GA-Nes line using anti-GA immunohistochemistry, we mainly detected poly-GA aggregates in neurons[34]. GFP fluorescence from cryosections revealed a more widespread expression of GFP-(GA)$_{175}$ in oligodendrocytes and astrocytes, which is consistent with the expression of Nestin-Cre also in glial progenitor cells. GA-Camk2a mice, which express poly-GA exclusively in excitatory neurons[43] and rNLS8 mice, with neuronal TDP-43ΔNLS expression, also express key DOL markers, hinting at a non-cell-autonomous induction mechanism. However, given the significant expression of poly-GA in oligodendrocytes in the GA-Nes mice used for most of this study, additional cell-autonomous effects are likely in this model.

Single-cell RNAseq analysis of treated GA-Nes mice suggests that cyclodextrin mainly rescues transgene-induced gene expression changes in oligodendrocytes, including cholesterol dysmetabolism, suggesting DOL conversion may depend on cholesterol or its derivatives. This parallels findings in cancer research, where lipid droplets have been implicated in inflammation and progression through eicosanoid signaling[68]. Interestingly, snRNAseq also shows strong upregulation of ApoD and the corresponding rescue by CD in GA-Nes mice, which mainly binds eicosanoids and lipids other than cholesterol[69].

DOLs have been observed in AD patients, but to a much lesser extent than in mouse models[14], possibly due to limitations in available antibodies for many secreted markers. Our findings may help to

identify DOL markers that are more widely expressed in human oligodendrocytes in neurodegeneration or demyelination.

## CD as a promising candidate for the therapy of neurodegenerative diseases

CD is commonly used as an excipient in pharmaceutical formulations to enhance the solubility and entrapment of various drugs. Its potential to sequester excess cholesterol has led to preclinical and clinical studies in neurodegeneration and atherosclerosis. In female GA-Nes mice, CD reduces NfL levels and cholesterol dysmetabolism in oligodendrocytes and partially rescues demyelination or improves myelination that occurs in the corpus callosum between P21 and P40. CD is beneficial in cellular and mouse models for Niemann-Pick type C disease, which is characterized by defective cholesterol export from lysosomes due to loss of NPC1 protein[25,70,71]. In the context of AD, CD administration has been associated with increased myelination[15] and decreased Aβ42 production[72]. In a mouse model of stroke, CD reduced the chronic inflammation and secondary neurodegeneration, thereby averting post-stroke cognitive decline[73].

Despite initial approval of compassionate use in selected NPC cases, the original CD formulation adrabetadex was later discontinued due to a negative risk/benefit ratio and lack of effect on Neurofilament light chain levels in the CSF[74]. However, a new CD formulation (Trappsol Cyclo) is currently tested in clinical trials for NPC and AD, with promising upregulation of several hydroxylated cholesterol species and reduction of Tau CSF levels as neurodegeneration marker in NPC patients[75]. Interestingly, NPC1 is essential for oligodendrocyte differentiation, and CD treatment partially restores myelination in NPC1 knockout mice[76]. While we detected consistent benefits in a *C9orf72* ALS model with daily CD treatment, weekly dosing had no obvious benefits in SOD1 ALS mice[77]. In addition, many biological studies using CD often do not report the degree of substitution of CD, although it greatly affects its colligative properties[78], and this could significantly affect treatment outcomes. Our gender-specific responses are consistent with data in NPC mice, although the mechanism remains elusive[39]. Interestingly, cholesterol dysmetabolism is more pronounced in male ALS patients, and testosterone affects myelination in mice[38]. The promyelinating effects of testosterone and differential cholesterol metabolism between the two genders may explain the dichotomy we observed in response to CD treatment, although further studies are needed.

Overall, our research underscores the significance of cholesterol dysmetabolism in C9orf72 ALS and suggests that alleviating CNS cholesterol overload could be a viable therapeutic strategy.

## Methods
### Animal experiments
All animal experiments were conducted in accordance with the German Animal Welfare Law and received approval from the Government of Upper Bavaria (licenses TV 55.2-2532. Vet_02-17-106 and TV 55.2-2532.Vet_03-17-68). All mice were housed in our pathogen-free animal facility in standard cages under a 12-h light/dark cycle with ad libitum access to water and food. Breeding and genotyping schemes of GA-Nes and GA-Camk2a mice followed those outlined in our previous publication[34]. The lines were maintained in a C57BL/6J background. The monogenic lines B6;C3-Tg(NEFH-tTA)8Vle/J (NEFH-tTA line 8, stock #025397) and B6;C3-Tg(tetO-TARDBP*) 4Vle/J (tetO-hTDP-43ΔNLS line 4, stock #014650) were obtained from the Jackson Laboratory (Bar Harbor, USA), backcrossed to C57BL/6J background, and intercrossed to generate bigenic regulatable NLS8 (rNLS8) animals[36]. Breeders as well as offspring mice were kept on a doxycycline diet (200 mg/kg, Ssniff, Germany) until the transgene was switched to standard chow lacking doxycycline (Ssniff, Germany).

For all symptomatic lines, mashed wet chow was provided on a petri dish on the cage floor, while the animals also had ad libitum access to normal food and water. Animals were scored daily during the experiment and were monitored for weight, basic awareness and neurological condition, agility, general well-being, and wound healing. While any of these criteria could necessitate euthanasia, GA-Nes mice reached their humane endpoint mainly due to 20% weight loss. Based on the uniform transgene expression, we used multiple brain regions for analysis to adhere to the 3R principles.

(2-Hydroxypropyl)-β-cyclodextrin (CAS #128446-35-5, here referred to as CD, Sigma–Aldrich H107-100G, Degree of Substitution = 5) was dissolved in 30 mM citric buffer (pH 5.0) in normal saline (vehicle) to achieve a 20% concentration. Starting from P21, all mice received daily subcutaneous injections of 2 g/kg CD solution or an equivalent volume of the vehicle. Mice in the survival cohort were treated until they reached the humane endpoint, whereas those in the fixed cohort were harvested at P40. Survival results were analyzed by R version 4.3 and survminer package version 0.4.9 (https://cran.r-project.org/web/packages/survminer/index.html).

Prior to tissue collection, intracardiac blood was collected and centrifuged at $2500 \times g$ for 10 min at 4 °C to isolate serum. Animals were perfused with ice-cold PBS prior to brain dissection, where one hemisphere was snap-frozen for biochemical investigation and the other prepared for immunohistochemistry.

GA-Nes mice in the survival experiment (both genders) were obtained from 17 litters, while the P40 cohort (only females were analyzed) was obtained from 30 litters. RNLS8 mice were obtained from three litters, while GA-Camk2a mice were obtained from two litters.

### Electron microscopy
Following perfusion and immersion fixation in 2.5% GA, 4% PFA, and 2 mM CaCl2 buffered in 0.1 M sodium cacodylate at pH 7.4 for 12 h, samples were stored for 3 days in 0.1 M sodium cacodylate buffer. Prior to further processing, $1 \times 1$ mm$^2$ pieces of the hippocampus were dissected from coronal sections with fine scalpels.

Heavy metal staining, dehydration, and resin infiltration were performed following a standard reduced osmium-thiocarbohydrazide-osmium (rOTO) en bloc staining protocol[79].

EM images were obtained on a Zeiss Crossbeam Gemini 340 SEM with a four-quadrant backscattered electron detector at 8 kV. After low-resolution imaging (9500 nm pixel size) to locate the sections, regions of interest were located at intermediate resolution (ranging from 50 to 200 nm/pixel). Regions of interest were acquired at 4 nm/pixel.

### Immunostaining
Paraffin sections were deparaffinized and rehydrated with xylene/alcohol washes. Heat-Induced Epitope Retrieval was performed in citrate buffer (pH 6) in a steam cooker. Frozen sections were brought to room temperature and subsequently rehydrated in PBS. For CA2+ oligodendrocyte staining in frozen sections, antigen retrieval was performed in citrate buffer (pH 6) at 80 °C. When using mouse IgG primary antibodies, ReadyProbes Mouse on Mouse IgG blocking solution (Invitrogen, R37621) was used, according to the manufacturer's specification. Subsequently, the sections were blocked and permeabilized with 5% FBS and 0.1% Triton-X-100 for 1 h at room temperature. Primary antibodies were prepared in blocking solution (2% FBS, 2% BSA, and 0.2% fish gelatin in PBS). Upon applying the primary antibody solution, slides were incubated in a humid chamber overnight at 4 °C, followed by two 5-minute washes in PBST (0.05% Tween-20 in PBS) and a final wash in PBS on the following day. After incubation of secondary antibodies for 1 h at room temperature, slides were washed as described before. Subsequently, the sections would be

incubated in a DAPI (4′,6-Diamidino-2-Phenylindole, Dihydrochloride, Invitrogen, D1306) solution for 20 min, followed by two PBS washes (5 min). Lastly, sections were coverslipped (No. 1.5H, 170 ± 5 μm) with ProLong Diamond Antifade Mountant (Invitrogen, P36970).

The following primary antibodies were used: Serpin A3N (R&D Systems, AF4709, 1:200), PLIN4 (Novus Bio, NBP2-13776, 1:200), GFAP (SySy, 173 006, 1:500), IBA1 (Abcam, ab283346, 1:400; and Fujifilm Wako, 019-19741, 1:500), CA2 (R&D Systems, MAB2184, 1:200), S100B (NovusBio, NB110-57478, 1:200), Dcx (Proteintech, 13925-1-AP, 1:100), NeuN (Abcam, AB104224, 1:200).

The following secondary antibodies were used: goat anti-chicken Alexa Fluor 647 (Invitrogen, A21449, 1:250), donkey anti-goat Alexa Fluor Plus 488 (Invitrogen, A32814, 1:250), donkey anti-goat Alex Fluor Plus 555 (Invitrogen, A32816, 1:250), donkey anti-rabbit Alex Fluor Plus 555 (Invitrogen, A32794, 1:250), goat anti-rat Alexa Fluor 647 (Invitrogen, A-21247, 1:250), donkey anti-mouse Alexa Fluor Plus 647 (Invitrogen, A32787, 1:250).

### Microscopy and image analysis
All fluorescence microscopy images in this study were captured using a Zeiss LSM710 confocal laser scanning microscope with ZEN 2010 software. Brightfield images were obtained with Leica DMi8 inverted microscope. To analyze images, Python 3.8 was used with the following packages: aicsimageio version 4.9.4 (https://github.com/AllenCellModeling/aicsimageio), numpy version 1.23.5[80], opencv-python version 4.6.0[81], scikit-image version 0.19.3[82], Cellpose[83], and pandas version 2.0.3. CZI files were read into Python using the AICSimageio library. Cellpose models were used to quantify cell count in confocal images as well as the number of myelinated axons in EM images. For Z-stacks, every z-plane was analyzed separately, and the results were averaged. To assess the staining area, images were thresholded (manually or with threshold_li from scikit-image, as appropriate). Further background areas consisting of very small noise were also removed using a size filter (scikit-image). Colocalization was assessed using numpy array overlap and labeled using scikit-image. The results of the analyzes were plotted with the ggplot2 library version 3.5.1 in R version 4.4.1.

### Immunoblotting
To extract proteins from tissue, frozen hindbrains were pulverized with a pestle and mortar. RIPA buffer (Serva, 39244.01) supplemented with 2% sodium dodecyl sulfate (SDS) and Halt protease inhibitor cocktail (Thermo Scientific, 78429) was added. Homogenization was done in Percellys 2 mL soft tissue tubes (Bertin Corp., P000912-LYSK0-A) with the Percellys homogenizer at 4 °C. Protein concentration was determined with BCA assay (Thermo Scientific, 23225).

For immunoblotting, Novex 10–20% Tricine gels (Invitrogen, EC66252BOX) were loaded with 20 μg of protein per well. Transfer was done using iBlot2® PVDF mini stacks (Invitrogen, IB24002) on an iBlot2 gel transfer device (Invitrogen, IB21001) using the default P0 program. Membranes were blocked with 0.2% iBlock (Invitrogen, T2015) in TBS Triton-X-100 for 1 h at room temperature, while shaking. Serpin A3N (R&D Systems, AF4709; 1:2000) and Calnexin (Enzo, ADI-SPA-860-F; 1:7000) antibodies and their respective anti-goat HRP and anti-rabbit HRP conjugated antibodies (1:5000) were used for western blotting. HRP-bound proteins were detected with Immobilon Forte Western HRP substrate (Millipore, WBLUF0500) using cytiva's Amersham ImageQuant 800 device.

### Immunoassays
To perform MSD immunoassays, V-PLEX Proinflammatory Panel 1 Mouse kit (MSD, K15048D-2) and V-PLEX Cytokine Panel 1 Mouse kit (MSD, K15245D-2) were used on hindbrain lysates. For the proinflammatory panel, 13 mg/mL of protein per well was taken, while 5 mg/mL of protein per well was needed for the cytokine panel. The assay was performed according to the manufacturer's protocol, and the plates were read with MSD's MESO QuickFlex SQ 120MM plate reader. Concentrations of the biomarkers were calculated in manufacturer's software, and the exported values within detection range were taken to R for visualization and statistical analysis, as described previously.

Serum NfL was measured with the NF-light Advantage Assay Kit (Quanterix, Cat. No. 103186) according to the manufacturer's protocol.

### Cholesterol and cholesteryl ester measurement
Midbrain homogenates were prepared at 100 mg/mL in PBS with 20 μg/mL BHT using Precellys Ceramic Beads at 4 °C. Homogenates were stored at −80 °C and then thawed to room temperature for lipid extraction. 50 μL of homogenate were transferred to glass Qsert vials containing 200 μL isopropanol with internal standards (UltimateSplash ONE, Avanti Polar Lipids). Samples were vortexed for 10 min at room temperature and then placed at −20 °C for 60 min. Samples were centrifuged for 20 min at 4000 × g at 10 °C. 150 μL supernatant was transferred to new glass Qsert vials. Samples were stored temporarily at −80 °C, then thawed, and 75 μL were diluted with 25 μL isopropanol prior to LC-MS analysis of cholesterol and cholesteryl esters. A Waters Acquity Premier UPLC system was used to inject 5 μL onto a Thermo Scientific Accucore C30 LC column (2.1 × 150 mm, 2.6 μm) at 45 °C at a constant flow rate of 0.26 mL/min with initial mobile phase composition 70% A (60% acetonitrile, 40% water, 0.1% formic acid, 10 mM ammonium formate) and 30% B (88% isopropanol, 10% acetonitrile, 2% water, 0.1% formic acid, 10 mM ammonium formate). Mobile phase B was increased to 43% at 2.00 min, to 55% at 2.10 min, to 65% at 12.00 min, to 85% at 18.00 min, to 100% at 20.00 min, held at 100% for 5.00 min, then equilibrated at 30% B for 5.00 min. Full MS scans in positive polarity were acquired on Thermo Scientific Q Exactive Plus mass spectrometer, with resolution setting 70,000 (at 200 m/z), AGC target 1e6, maximum injection time 256 ms, with scan range 365–380 m/z from 0 to 16 min and 630–780 m/z from 16 to 30 min. Peak areas for cholesterol and cholesterol esters were extracted from the raw data using Thermo Scientific Xcalibur software.

### snRNAseq nuclei isolation and library preparation
Nuclei from P40 female hippocampi with acceptable RNA quality RNA Integrity Number (RIN) >7, with Agilent RNA 6000 Nano kit (Agilent, 50671511), were isolated according to Frankenstein protocol for nuclei isolation from fresh and frozen tissue[84]. Nuclei were sorted via fluorescence-activated nuclei sorting (FANS) by a DAPI+ gate and two further gates of NeuN-positive and NeuN-negative nuclei (anti-NeuN antibody from Abcam, ab190195) to obtain 5000 neurons and 5000 glia per sample, respectively.

The nuclei were processed with Chromium Next GEM Single Cell 3′ Kit v3.1 (10x Genomics, 1000268) generate the cDNA libraries for this study. Gel Bead-in-Emulsions (GEMs) were generated with chip G (10x Genomics, 1000120) on the microfluidic platform 10x Genomics Chromium Single Cell Controller. Next, the samples were uniquely barcoded to distinguish the mRNAs (each having unique molecular identifier (UMI)) from different nuclei.

The cDNA was extracted with Dynabeads MyOne SILANE beads (Invitrogen, 370–12 D) and amplified by PCR. The PCR product was purified with SPRIselect reagent (Beckman Coulter, Cat #B23318). Subsequently, the 3′ gene expression library was constructed with primers for Illumina NGS platform and Dual Index Kit TT set A (10x Genomics, 1000215), which allows individual sample identification.

The pooled libraries were sequenced on Illumina's NovaSeq6000 system using S4 flow cell type (500 M reads/sample) with a 2 × 150 bp paired-end read length.

### snRNAseq data analysis
The raw sequencing data were initially processed using Cell Ranger software from 10x Genomics to align reads to the mouse reference

genome, followed by the creation of a Seurat object. A series of quality control measures were performed on the data where doublet score was calculated on a sample level. Then clustering was performed with high resolution, and clusters with more than 20% doublets were completely removed. Next, clusters with low-quality cells (mitochondrial gene expression, too many/few genes/UMIs) were also fully removed. Subsequently, we performed more rigorous quality control, where we deleted the individual cells with a higher-than-average doublet score or mitochondrial content using Seurat version 4.3.0[85] in R 4.2.0.

Using Seurat's built-in functions, the Seurat object was normalized by a scale factor of 10,000, and variable features were found using 3000 features with the "vst" selection method. The data was then scaled, and principal component analysis (PCA) was run on it with 25 dimensions (minimum 15 necessary as per elbow plot). Next, the data was integrated using Harmony version 0.1.1[86]. The Seurat object subsequently underwent dimensionality reduction by UMAP with 45 dimensions (as informed by the elbow plot) and the reduction parameter set to 'harmony'. "FindNeighbors" function was then used to prepare the data for clustering, with 45 dimensions. To achieve optimal clustering, we tried multiple resolutions in the function "FindClusters". Based on analysis of these clusters with clustree version 0.5.0[87] and manual inspection of the UMAPs, we chose a resolution of 0.45 as the best compromise.

To annotate the clusters, "FindAllMarkers" function from Seurat was used, and the outputs were sorted. Some clusters were identified by known markers, such as GFAP, as a marker for astrocytes. For other clusters, the top markers were fed into PanglaoDB[88] to have a more informative and less biased clustering. Clusters that identified dentate gyrus (DG) neurons were identified with Janelia's Hipposeq[89] based on their top markers.

To perform pseudobulk analysis, the data were converted using SingleCellExperiment package version 1.20.0[90]. The complete expression data are shown in Supplementary Data 3. Aggregated count data were fed to DESeq2 package version 1.36.0[91] to obtain differentially expressed genes and generate PCA plots. All FeaturePlots and DotPlots were generated via Seurat's built-in functions.

## iPSC derived neuron model

Human neurons were generated by doxycycline-mediated induction of NGN2 expression in small molecule neural precursor cells (smNPC) from a healthy female donor as previously described (GM23280, iPSC line obtained from the Coriell Institute)[92,93]. The smNPCs are maintained in expansion medium containing N2B27 medium supplemented with CHIR99021 (StemCell Technologies, 100–1042; 3 μM), PMA (0.5 μM), and ascorbic acid (Sigma, A8960; 64 mg/L). For treatment, smNPCs are differentiated into iNeurons through NGN2 expression induced by doxycycline (2.5 μg/mL). After induction, smNPCs differentiate into iNeurons within 5 days, demonstrating robust neuronal identity in this model, with MAP2 as a pan-neuronal marker and BRN2 as a marker for upper cortical layer neurons[93,94], which was further confirmed by transcriptomics (GEO GSE285209).

iNeurons are transduced with (GA)$_{149}$-GFP and GFP lentiviral constructs on day 7, media change to remove excess virus on day 8, and CD (2.5 mg/mL) treatments were initiated at day 10. CD was added to the media at the same concentration every other day. On day 19, RNA was isolated using Qiagen RNeasy Micro Kit according to the manufacturer's instructions. The transcriptome analysis was performed at BGI (China). After cDNA library preparation, 100 bp pair-ended sequencing was performed on a DNBSEQ platform at a depth of ~20 million read pairs per sample. Reads were aligned to the human genome (GRCh38.p14) at BGI, and differential gene expression was analyzed using SummarizeOverlaps and DESeq2[91]. The complete expression data are shown in Supplementary Data 4.

## Bulk RNAseq and analysis

We reanalyzed several previously published transcriptomics datasets for GA-Nes mice[34], GA-CFP mice (GEO GSE138413)[37], and rNLS8 mice (GEO GSE233669)[51]. To replicate the GA-CFP (GSE289729) and rNLS8 (GSE290262) data, we used the control groups from ongoing treatment studies. Briefly, total RNA was extracted from mouse brain cortex with the RNeasy Plus Mini Kit (Qiagen) according to the manufacturer's instructions. Quality control was performed, and RNA samples with RIN values above 7 (Agilent) were sent to BGI Genomics (China) for transcriptome analysis. Data were aligned to the mouse genome (GRCm38), and differential gene expression was analyzed using DESeq2. For the GA-Camk2a mouse line we used Prime-seq[95] from multiple brain regions for a more extensive study. The relevant data can be found at GEO with accession GSE291440. Data were aligned to the mouse genome (GRCm39), and differential gene expression was analyzed using DESeq2. For the human ALS transcriptomics data, we downloaded all publicly available RNAseq data from TargetALS (June 2024). We included all spinal cord samples from patients with ALS and controls without CNS comorbidity and with available C9orf72 genotype information. Reads were aligned to GRCh38 using STAR aligner v2.4.2a from the New York Genome Center and further processed by us using SummarizeOverlaps in R. Count tables were analyzed using DESeq2[91] in a linear model adjusting the diagnosis for sex and anatomical subregions (cervical: 19 controls, 131 ALS, 28 C9orf72 ALS; thoracic: 9 controls, 46 ALS, 7 C9orf72; lumbar: 16 controls, 122 ALS, 24 C9orf72 ALS). Analyzing two subgroups (before and after June 2020) gave the same results (data not shown).

## Statistics and reproducibility

Statistical analyzes were performed using R version 4.3. Statistical significance was determined using Welch's t-test or Mann–Whitney U-Test for pairwise comparison or using one-way ANOVA for multiple comparisons followed by Tukey's HSD posthoc or Fisher's LSD. All analyzes were two-tailed. Tukey's HSD posthoc inherently corrects for multiple comparisons. The survival curves were generated with the Survminer package in R, and the statistical comparison was conducted using Log-Rank test. Experimental group sizes (n) and p value significance levels are reported in the figures, their legends, and the Source Data file. p values less than 0.05 were considered to be statistically significant. In all the boxplots, the bounds of the box span from 25% to 75% percentile (first and third quartiles), and the whiskers represent the most extreme data points that are within 1.5 times the interquartile range (IQR) of those quartiles. The line in the middle of boxplots represents the median. In addition, individual data points are plotted. Source data are provided for all the underlying raw data, statistics, and blots.

After observing beneficial effects of CD in a pilot study, we generated three additional cohorts for replication and tissue collection for biochemistry, histology, and EM analysis, with reproducible results. Treatment with CD or vehicle was randomized using a random number generator. snRNAseq and its initial analysis were blinded. The genotype of GA-Nes mice could not be concealed from researchers due to obvious pathology. The subsequent wet lab experiments (e.g., staining) were effectively blinded.

## Reporting summary

Further information on research design is available in the Nature Portfolio Reporting Summary linked to this article.

# Data availability

Source Data is provided as a Source Data file including raw data and underlying statistical analyzes of figures. SNRNAseq data can be accessed under GEO accession number GSE262778. Bulk RNAseq data iPSC-derived neurons (GSE285209), GA-CFP mice (GSE289729), rNLS8

(GSE290262), and GA-Camk2a mice (GSE291440) are available at NCBI GEO. Data is available from the corresponding author upon request. Source data are provided with this paper.

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

## Acknowledgements

We acknowledge funding from the Deutsche Forschungsgemeinschaft (DFG, German Research Foundation) SyNergy EXC 2145—390857198 (D.E., M.S., E.B., G.K., Q.Z.), the Thierry Latran foundation (DE), a Sanofi iAward (DE), European Research Council (ERC) Starting Grant (C9-Immunity, 101117710, QZ) and the Horizon Europe Framework Program (HORIZON) under grant agreement 101057649 (GA-VAX) (D.E. and Q.Z.). S.F. was supported by the DFG Research Infrastructure NGS_CC (project 407495230) as part of the Next Generation Sequencing Competence Network (project 423957469). NGS analyzes were carried out at the Competence Centre for Genomic Analysis (Kiel). We thank Meike Michaelsen for cryostat sectioning, Martina Fetting for input on the electron microscopy, and Christian Haass for providing the Simoa platform for serum NfL measurements. We thank Irina Dudanova, Christian Haass, Timothy Hammond, and Veit Hornung for helpful discussions. We thank TargetALS for providing RNAseq data from ALS patients and controls.

## Author contributions

A.R., N.H., D.O., and D.E. conceived the project. A.R., Z.I.G., N.H., D.O., E.B., O.G., M.S., S.L., and D.E. designed research. A.R., V.K.-J., Z.I.G., Q. Zeng, G.K., H.B.I., T.K., L.R.P., S.F., K.D.L., A.J. performed the experiments with support from J. Koppenbrink, J. Knogler, D.F., B.N., E.K., G.G., A.D., C.Y., Q. Zhou. A.R., F.B., G.K., T.K., L.R.P., E.B., A.J., T.A., and D.E. analyzed data. The work was supervised by W.E., M.S., S.L., and D.E. The manuscript was written by D.E. and A.R. with contributions from all authors.

## Funding

## Competing interests

L.R.P., N.H., and D.O. are employees of Sanofi, but Sanofi is not actively developing CD-based therapeutics. The remaining authors declare no competing interests.

## Additional information

**Ali Rezaei** [1,2,3], **Virág Kocsis-Jutka**[1,17], **Zeynep I. Gunes** [2,3,4,5,17], **Qing Zeng** [1,3,17], **Georg Kislinger**[1,2], **Franz Bauernschmitt**[2,4,5], **Huseyin Berkcan Isilgan** [1], **Laura R. Parisi**[6], **Tuğberk Kaya**[1,2,7], **Sören Franzenburg** [8], **Jonas Koppenbrink**[1], **Julia Knogler**[1], **Thomas Arzberger**[9,10], **Daniel Farny**[1], **Brigitte Nuscher**[11], **Eszter Katona**[1,2,3], **Ashutosh Dhingra**[12], **Chao Yang**[1], **Garyfallia Gouna** [1,3,13], **Katherine D. LaClair** [1], **Aleksandar Janjic**[14], **Wolfgang Enard**[14], **Qihui Zhou** [1,2], **Nellwyn Hagan**[6], **Dimitry Ofengeim** [6], **Eduardo Beltrán** [2,4,5], **Ozgun Gokce**[2,3,7,15], **Mikael Simons** [1,2,3,13], **Sabine Liebscher** [2,3,4,5,16] & **Dieter Edbauer** [1,2,3] ✉

[1]German Center for Neurodegenerative Diseases (DZNE), Munich, Germany. [2]Munich Cluster of Systems Neurology (SyNergy), Munich, Germany. [3]Ludwig-Maximilians-Universität (LMU) Munich, Graduate School of Systemic Neurosciences (GSN), Munich, Germany. [4]Institute of Clinical Neuroimmunology, Klinikum der Universität München, Ludwig Maximilians University Munich, Munich, Germany. [5]Biomedical Center, Ludwig Maximilians University Munich, Munich, Germany. [6]Sanofi, Rare and Neurologic Diseases, Cambridge, MA, USA. [7]Department of Neurodegenerative Diseases and Geriatric Psychiatry, University Hospital Bonn, Bonn, Germany. [8]Institute of Clinical Molecular Biology, Kiel University, Kiel, Germany. [9]Center for Neuropathology and Prion Research, University Hospital, LMU Munich, Munich, Germany. [10]Department of Psychiatry and Psychotherapy, University Hospital, LMU Munich, Munich, Germany. [11]Chair of Metabolic Biochemistry, Biomedical Center (BMC), Faculty of Medicine, Ludwig-Maximilians-Universität Munich,

Munich, Germany. [12]German Center for Neurodegenerative Diseases (DZNE), Tübingen, Germany. [13]Institute of Neuronal Cell Biology, Technical University Munich, Munich, Germany. [14]Anthropology and Human Genomics, Faculty of Biology, Ludwig-Maximilians Universität München, Munich, Germany. [15]Institute for Stroke and Dementia Research, University Hospital of Munich, LMU Munich, Munich, Germany. [16]Institute of Neurobiochemistry, Medical University of Innsbruck, Innsbruck, Austria. [17]These authors contributed equally: Virág Kocsis-Jutka, Zeynep I. Gunes, Qing Zeng. ✉e-mail: dieter.edbauer@dzne.de

