## [Transparent Peer Review file · Nature Communications]

Correction of dysregulated lipid metabolism normalizes gene expression in oligodendrocytes and prolongs lifespan in female poly-GA C9orf72 mice

Corresponding Author: Professor Dieter Edbauer

Version 0:

Reviewer comments:

Reviewer #1

(Remarks to the Author)

In this study, the authors used poly-GA expressing mice as a fast-progressing model of C9orf72-ALS/FTD. Although it is not a perfect model by any means, the insight obtained here could be still informative. They found that a cholesterol-binding drug, 2-hydroxypropyl- β -cyclodextrin (CD), normalized cholesteryl ester levels and increased lifespan in female but not male GA mice, without affecting poly-GA aggregates. A single-nucleotide RNA-seq analysis showed that oligodendrocytes are the main cell type affected by CD. CD treatment greatly restored myelin gene expression and rescued the disease-associated defects in the oligodendrocyte response. These results suggest a new therapeutic strategy, although validation in other mouse models of C9orf72-ALS/FTD, such as BAC transgenic mice that show disease phenotypes, will be needed in the future.

1. Please state in the Abstract that “CD treatment prolonged lifespan in female but not male GA-Nes mice.”
2. The terms “poly-GA mice” or “GA-Nes mice” are used interchangeably. Please use one term consistently throughout the manuscript.
3. On page 3, the authors state that “Vehicle-treated mice reached the endpoint around 42 days of age (Fig. 1B)”. If this is true for both male and female mice, they should say “Vehicle-treated male or female mice reached the endpoint around 42 days of age (Fig. 1B, 1C).”
4. In the text, they use capital letters to refer figure panels. But in the figures, the panels are labeled with lowercase letters. Please use one consistently.
5. On page 4, the authors state that “CD treatment had no obvious effect on the widespread poly-GA expression in poly-GA mice.” This conclusion should be backed up by a more quantitative western blot analysis.
6. On page 4, line 119, the authors suddenly mention GFP-(GA)¹⁷⁵. The authors should describe at the beginning of the Results section exactly what mouse model they are using here.
7. Additional immunostaining experiments are needed in Figure 1E. Previously they reported that in nestin-cre mice, poly-GA is predominantly expressed in neurons. But Figure 1E shows mostly cells that appear to be microglia. How many cells in this image are neurons? Co-immunostaining with neuron- or microglia-specific markers should be performed.
8. The authors claim that they studied a mouse model of ALS but used the hippocampus for snRNA-seq analysis. The experiment should be repeated with motor cortex or spinal cord tissues. If it is too much work to redo snRNA-seq, then a better justification for using hippocampus should be provided.
9. Seven subtypes of microglia should be identified by single-cell analysis. It is unclear why they clustered microglia into only two groups in their study.

10. The results from the GA-CaMK2a model should be removed, as they only used one mouse per genotype. Alternatively, if they wish to draw a solid conclusion, they should study additional mice.

Reviewer #2

(Remarks to the Author)

In this manuscript, Rezaei and colleagues describe cholesterol alterations in ALS and focus on the oligodendrocyte population to

The study is potentially informative but has major limitations such as the use of a very aggressive and artificial mouse model and reduced translation to human (the RNAseq in Fig. 1 is helpful as an entry point but more human-based data are needed to increase the relevance of the findings).

My major points are listed below:

- How did the authors compare the different RNAseq in Fig. 1? How do they deal with batch effect and heterogeneity of the datasets when integrating them? More information and quality controls must be provided.
- The human RNAseq from the van Rheenen publication has a heterogeneous genetic background, the authors should highlight C9orf72 patients or perform a second layer of analysis with C9-mutant individuals.
- The altered expression of the key genes identified by the authors must be confirmed in independent experiments both in mice and human samples.
- CD treatment should be performed in other ALS models, such as the TDP43 used in for the transcriptome in Fig. 1, but information on other DPR species must also be provided.
- Since the treatment is more effective in females, analysis of the transcriptome by separating the sexes should additionally be provided to justify these differences. Also, this must be as well compared to human data
- Fig. 1E is not informative in its present form. This might be due to the low quality of the Figure after downloading from the online system but nevertheless the authors must provide higher-resolution imaging (such as STED) must be performed and quantified to better characterize the GA localization, especially because the authors focus later on oligodendrocytes.
- On the same line, the authors claim based on the SN RNAseq in Fig. 2 that GA expression results in "depletion of astrocytes", this should be better contextualized in relation to the data in Fig. 1E.
- Fig. 2 b-c-d: I have major concerns on the validity of the PCs used for the analysis. How many PCs were identified in total? I am doubting if PC2 is really informative since there is a dramatic drop of % from PC1 (between 80 and 90% in the 3 PCAs) and PC2 (4 to 7%). This might represent noise in the dataset and must be better characterized. The shift of the samples upon treatment is indeed explained by PC2 and the authors must provide more detailed analysis on this. Euclidian distance should be calculated and PC loading provided to identify the genes responsible for this shift. Do the genes that mostly drive this transcriptional change pass quality control for expression abundance?
- As a supplemental information, the same PC analysis must be provided with the entire dataset and not only with the top 500 genes.
- Are the genes "rescued" by CD sorted according only to LFC or also adj P value?
- The effect of CD on myelination is interesting: does it apply only to the corpus callosum or also to the white matter of the spinal cord? Did the authors observe amelioration in the motor phenotype upon treatment?
- Expressing GA only in neurons trigger a similar phenotype, which the authors address as "non-cell autonomous" effect. This is rather a superficial explanation and the mechanism needs to be better elucidated to clarify why suffering neurons impact in this way myelinating cells.
- The main cholesterol-related alterations observed in the polyGA mice must be investigated in hiPSC-derived neurons or organoids. Which of the alterations are conserved? Is CD treatment beneficial also in human models? This might also help in addressing the previous point.

Reviewer #3

(Remarks to the Author)

Rezaei et al., presents interesting findings on the beneficial effects of the cholesterol binding drug CD in improving survival of female Nestin Cre-poly GA mice – a mouse model for C9ORF72 related ALS/FTD. The authors have used existing transcriptomics data set from various ALS animal models and patient autopsy to set up a hypothesis that decreased cholesterol biosynthesis. To test this hypothesis, they have administered CD to the poly GA mice which showed reversed expression changes in mature oligodendrocytes and neuroblasts in the hippocampus of young adult transgenic mice that exhibit myelin defects. CD drug has been known to target cholesterol polyesters and there is some evidence in the article to support this with the drug normalising elevated cholesterol polyesters in the midbrain of young adult Nes Cre-poly GA mice. The authors also highlight a cholesterol ester binding protein, Plin4, to be specifically upregulated in oligodendrocytes in transgenic mice with expression partially rescued with CD, advocating for lipid metabolism as a viable therapeutic target. Overall, this study is of relevance in understanding ALS disease pathology, redirecting focus back to oligodendrocytes, a largely understudied cell type. Moreover, the specific effect of the CD drug on oligodendrocyte maturation and subsequent improvement in survival rates in poly GA transgenic mice is noteworthy.

The authors place much importance on the cholesterol biosynthesis pathway being affected in oligodendrocytes leading to myelin defects via a non-cell autonomous mechanism in the CamK2-poly GA mice. Further characterisation is needed to support some aspects of this hypothesis. The manuscript presents limited data on cholesterol synthesis gene expression changes in oligodendrocytes in transgenic mice (with and without CD treatment). The myelin changes in transgenic mice need further studies and the inclusion of CD treated transgenic mice. Moreover, it would be important to elucidate if the poly

GA inclusions are present in oligodendrocytes as well as neurons in their primary Nestin Cre driven mouse model to shed light on disease aetiology and their potential key finding of oligodendrocytes/cholesterol pathway as a target for therapeutics.

Specific comments

- 1) Fig 1: The drug CD specifically improves survival of female transgenic mice, normalising NfL levels in the blood. Authors should also publish NfL levels from the male mice as well to shed light on the gender specific role of CD in this mouse model.
- 2) Fig 1: Although measurement of NfL levels, as a surrogate for neurodegeneration, is commendable and will be helpful in translational studies, noting the discovery nature of this study the authors should undertake conventional histopathology studies to understand the cellular differences neuronal count, astrocyte/microglial reactivity/oligodendrocytes, to associate which cells contribute to the observed survival benefit.
- 3) Fig 1: The lifespan experiments have used weight loss as a proxy for humane endpoints in Fig1C. It would be good if the authors could depict the weights as a graph with information from each animal, in addition to the survival plots since this would be a more accurate representation of the effect of CD on survival of transgenic mice. Moreover, the survival plots show that only about 25-50% of the female mice are significantly improved with CD treatment. The authors should discuss these findings and comment on heterogeneity in the survival, detailed histopathological analysis might help.
- 4) Fig 1: The authors allude towards poly GA inclusions being present in both neurons and glia, however, data to support this claim is missing in the current manuscript. They could consider including immunohistochemistry for glial cell type markers co-labelled with CA-GFP with subsequent quantifications to understand if aggregate pathology is also originating in oligodendrocytes and driving myelin defects directly via a cell-autonomous mechanism.
- 5) Fig 2: the authors present elegant snRNAseq data from different cell types showing that mature oligodendrocytes and neuroblasts have the most transcriptional changes with CD treatment normalising several key myelin genes. However, there is no specific dataset examining the status of cholesterol biosynthesis genes in each cell type. Given that the foundation of the study examines the cholesterol pathway, the authors should consider including this dataset, particularly in oligodendrocytes. The authors also remark on the relative depletion of astrocytes, neuroblasts and oligodendrocytes in the transgenic mice from the snRNAseq data. Given the non-empirical nature of FANS/snRNAseq, it would be good to support these claims with immunohistochemistry for cell type specific markers in the transgenic/CD treated mice.
- 6) Fig 3D: For the Serpin A3N immunoblotting experiment the authors have used pooled lysates from number of brains. This masks the biological variability, thus, authors should consider repeating this experiment keeping animals separate with subsequent quantifications to reflect experimental variability and robustness of the data.
- 7) Given the marked gene expression changes in oligodendrocytes, the authors briefly examine myelin changes in transgenic mice. This dataset is limited and potentially needs further characterisation to assess if myelination is impaired in transgenic mice or if it is a type of axonopathy that is observed. The authors could include quantifications such as G-ratio and/or axon diameter measurements. The images look like there is a drastic loss of myelinated axons in the corpus callosum. How does this compare with the hippocampus wherein the snRNAseq was performed? In general, the manuscript examines several regions in different experiments (midbrain, hindbrain, hippocampus, and cortex/corpus callosum). It would be worth discussing region specific changes as well as maintaining consistency within brain regions for gene expression and immunoblotting/immunohistochemistry.
- 8) Myelin analysis in CD treated mice is missing and would be key to understanding if the drug reverses only oligodendrocyte gene expression or if this is also translated to oligodendrocyte function being restored. It could be helpful to perform analysis at earlier timepoints to better understand if the myelin phenotypes observed is due to developmental myelination or demyelination. For eg. at P21 when drug is administered, is myelin comparable in the genotypes as this would speak towards a rescue of demyelination by CD.
- 9) The final section of the manuscript examines the expression of the cholesterol specific gene Plin4. The specific upregulation of Plin4 in only mature oligodendrocytes is striking and interesting. If the authors could examine Plin4 expression in human postmortem ALS tissue or in a human stem cell model this would be very helpful in understanding the importance of Plin4/lipid metabolism as a therapeutic target in ALS human pathology.
- 10) The Camk2a-Cre data alludes largely towards unpublished data from another manuscript in preparation. The data included in this manuscript is quite preliminary with limited evidence supporting the conclusions. Moreover, the scRNAseq experiment is underpowered with only one biological replicate. The authors should consider including further characterisation of the Camk2aCre model, particularly in relation to CD treatment or completely removing this dataset and soften the discussion around non-cell autonomous role. This dataset is quite crucial as it can address if poly GA pathology and subsequent oligodendrocyte dysfunction occurs via cell-autonomous or non-cell autonomous mechanisms.
- 11) As a more general comment, the authors should include additional details in methods/figure legends mentioning number of litters and if error bars are values from individual animals/sections.

(Remarks to the Author)

I co-reviewed this manuscript with one of the reviewers who provided the listed reports. This is part of the Nature Communications initiative to facilitate training in peer review and to provide appropriate recognition for Early Career Researchers who co-review manuscripts

Version 1:

Reviewer comments:

Reviewer #1

(Remarks to the Author)

The authors did a good job to address all my concerns raised earlier. Thus I support its publication now. Thanks.

Reviewer #2

(Remarks to the Author)

The authors have efficiently and extensively addressed, either experimentally or in the discussion, my comments. The manuscript in my opinion has been strongly improved and should now be considered for publication in Nature Communications.

Reviewer #3

(Remarks to the Author)

The authors have addressed most of the key issues raised previously.

Few minor suggestions,

1) It is unclear how the extended survival in a subset of female mice is achieved. Authors observed CD treatment only increased myelination and did not alter other cell types. Did the authors perform pathological characterisation only in female mice or both sexes were included? It would be necessary to clarify in the text. If the study combines both sexes, this suggests improvement in myelination alone is not sufficient to increase survival and perhaps other mechanisms are involved. Would suggest to discuss this.

2) Human iPSC neuron study is not fully characterised, i.e. identity of neurons, efficiency of poly-GA formation, suggest removing this data as it doesn't add further knowledge to the study.

Reviewer #4

(Remarks to the Author)

Summary of changes in the revised manuscript

We extend our sincere gratitude to the reviewers for their constructive and insightful feedback, which has significantly enhanced the rigor and clarity of our study. Below is a brief summary of the key changes made in response to your suggestions:

- Extended Figure 1a: We replicated the cholesterol dysmetabolism in additional RNAseq datasets from independent cohorts of GA-CFP and rNLS8 mice, both models with neuron-specific aggregate expression. Analysis of an expanded patient cohort revealed more severe phenotypes in male ALS patients but no differences between sporadic and C9orf72 ALS cases.
- New Figure 1d: NfL levels in male GA-Nes were not affected by CD treatment, which is consistent with the lack of survival benefits in males.
- New Figure S1: We provide individual body weight curves for the survival cohort in Figure 1b/c to showcase the beneficial effect of CD in females GA-Nes mice as indicated by reduced weight loss.
- New Figure 1f/g: We have replaced the original images with higher quality images showing neuronal and glial expression of poly-GA as requested. Quantitative analysis confirms that CD treatment does not significantly affect poly-GA levels in GA-Nes mice. Additional high-resolution images are shown in the new Figure S2.
- New Figure S2: We analyzed expression of poly-GA in different glia cell types in GA-Nes mice show a previously underappreciated expression of the transgene in oligodendrocytes and astrocytes.
- New Figure S6: Quantitative cell-type analysis shows an increased number of microglia and astrocytes in GA-Nes mice, while neuron and neuroblast numbers were decreased. CD treatment did not significantly rescue these changes, although there was a trend for higher neuron number and reduced astrocyte number in CD treated mice.
- New Figure S9: Cell-type-specific pathway analysis shows that cholesterol export and storage pathways were mostly induced in oligodendrocytes of GA-Nes mice and less consistently in microglia and astrocytes. CD treatment had the greatest rescue effect on oligodendrocytes.
- Figure 1a/3a: Additional bulk RNAseq data show cholesterol dysmetabolism and DOL induction also in the spinal cord of GA-Nes mice, although less severe than in the hippocampus and neocortex.
- Extended Figure 3d/e: We replaced the pooled analysis of Serpina3n by western blots of individual animals. Densitometric analysis shows significant induction by SerpinA3N in GA-Nes mice, which was rescued by CD treatment.
- New Figure 4 and S13b/c: We greatly extended the EM analysis of myelination in GA-Nes mice. Quantitative analysis shows a loss of myelinated axons in GA-Nes, which is partially rescued by CD treatment. Importantly, myelination in GA-Nes mice and control littermates was not significantly different at the start of the CD treatment (new Figure S13b/c). This suggests that GA-Nes mice undergo active demyelination or myelinate less between P21 and P40, which is ameliorated by CD treatment.
- New Figure S13d: We analyzed cholesterol metabolism upon poly-GA expression and CD treatment in human iPSC-derived neurons. In line with the GA-Nes data (new Figure S9), poly-GA had little of effect on neuronal cholesterol expression, but CD caused compensatory activation of cholesterol biosynthesis.

- We redid the PCA analysis with all genes instead of the top 500 variable genes as requested (revised Figure 2b-d and S7) and provide PC loadings and Euclidian distances between groups and individual animals as requested (Table S2).
- As requested by reviewers #1 and #3, we removed the underpowered scRNAseq dataset in GA-Camk2a mice with pure neuronal poly-GA expression. To still support the conclusion, we extended the characterization of DOLs in this mouse model and the rNLS8 TDP-43 mouse model using bulk RNAseq and immunofluorescence (new Figure 6a-e). Nevertheless, we have revised the discussion to soften our claim about non-cell-autonomous effects of poly-GA expression on DOL induction.
- In addition, we plotted all individual datapoint in the boxplots throughout the manuscript as mandated by the journal policy.

Together, these revisions provide a more comprehensive and robust exploration of CD's effects on cholesterol metabolism and myelination while addressing all key points raised by the reviewers.

Point by point response:

Reviewer #1:

In this study, the authors used poly-GA expressing mice as a fast-progressing model of C9orf72-ALS/FTD. Although it is not a perfect model by any means, the insight obtained here could be still informative. They found that a cholesterol-binding drug, 2-hydroxypropyl- β -cyclodextrin (CD), normalized cholesteryl ester levels and increased lifespan in female but not male GA mice, without affecting poly-GA aggregates. A single-nucleotide RNA-seq analysis showed that oligodendrocytes are the main cell type affected by CD. CD treatment greatly restored myelin gene expression and rescued the disease-associated defects in the oligodendrocyte response. These results suggest a new therapeutic strategy, although validation in other mouse models of C9orf72-ALS/FTD, such as BAC transgenic mice that show disease phenotypes, will be needed in the future.

We thank the reviewer for the interest in our study and the constructive feedback. We agree that future studies in BAC transgenic mice might be rewarding. We extended validation of DOL induction in GA-Camk2a mice and added new data in a TDP-43 mouse model.

1. Please state in the Abstract that "CD treatment prolonged lifespan in female but not male GA-Nes mice."

We agree and we have revised the abstract as suggested to ensure clarity and accurate representation of our findings.

2. The terms "poly-GA mice" or "GA-Nes mice" are used interchangeably. Please use one term consistently throughout the manuscript.

We agree with the reviewer that consistent terminology is essential for clarity. We have standardized the use of "GA-Nes mice" throughout the manuscript.

3. On page 3, the authors state that “Vehicle-treated mice reached the endpoint around 42 days of age (Fig. 1B)”. If this is true for both male and female mice, they should say “Vehicle-treated male or female mice reached the endpoint around 42 days of age (Fig. 1B, 1C).”

We have revised the text to state: " Vehicle-treated male and female mice reached the endpoint at a median age of 41 and 43 days, respectively, in line with our initial characterization of this line (Fig. 1b/c)."

4. In the text, they use capital letters to refer figure panels. But in the figures, the panels are labeled with lowercase letters. Please use one consistently.

To maintain consistency across the manuscript, all figure panel labels have been updated to lowercase letters in both the text and figure captions.

5. On page 4, the authors state that “CD treatment had no obvious effect on the widespread poly-GA expression in poly-GA mice.” This conclusion should be backed up by a more quantitative western blot analysis.

We appreciate the reviewer's suggestion and have quantified poly-GA expression by immunofluorescence, which confirms that CD treatment does not significantly alter poly-GA levels (new Figure 1g). We chose immunofluorescence rather than Western blotting because poly-GA tends to form aggregates that form a high molecular weight smear across the gel and/or get trapped in the loading pockets during SDS-PAGE, making quantification unreliable.

6. On page 4, line 119, the authors suddenly mention GFP-(GA)175. The authors should describe at the beginning of the Results section exactly what mouse model they are using here.

To address this point, we now mention the exact transgene in the GA-Nes line at the end of the introduction and specified the mouse model (“GA-Nes”) in the referenced section.

7. Additional immunostaining experiments are needed in Figure 1E. Previously they reported that in nestin-cre mice, poly-GA is predominantly expressed in neurons. But Figure 1E shows mostly cells that appear to be microglia. How many cells in this image are neurons? Co-immunostaining with neuron- or microglia-specific markers should be performed.

We agree with the reviewer that a clearer characterization of cell types expressing poly-GA in the GA-Nes model is critical. The original images were suboptimal as they focused predominantly on the glial component. In response, we have replaced the original Figure 1e with higher-quality images of the dentate gyrus, including NeuN co-staining, which confirms strong neuronal expression of the transgene while still showing the glial component (new Figure 1f). Quantitative analysis demonstrates that CD treatment does not significantly alter total poly-GA expression (new Figure 1g).

To further clarify the glial expression of the GFP-(GA)₁₇₅ transgene, we have added co-immunostaining experiments with glial markers (new Figure S2). Our analysis shows colocalization of Iba1-positive microglia with small poly-GA dots, which may reflect either phagocytic uptake or low levels of endogenous expression. By contrast, most of S100 β -positive astrocytes exhibit high levels of poly-GA expression, while most of oligodendrocytes also show small areas of poly-GA expression. These findings highlight a previously underappreciated widespread glial expression of the transgene in the GA-Nes model, which was not apparent to this extent using immunohistochemistry or RNAscope (LaClair et al, Acta Neuropath 2020).

Importantly, we confirmed cholesterol dysmetabolism in the GA-Camk2a mice (new Figure 5i) and a second cohort of GA-CFP mice (extended Figure 1a), where neuron-specific promoters exhibit Camk2a-Cre or the Thy1 promoter drive neuronal poly-GA expression (Schludi et al, Acta Neuropathol 2017). Moreover, we confirmed induction of DOL-like cells in GA-Camk2a mice and the newly analyzed rNLS8 mouse model with predominant neuronal transgene expression driven by the neurofilament heavy chain promoter (new Figure 6b-f).

Finally, given the widespread expression of C9orf72 in neurons and glia (as shown in the Human Protein Atlas: <https://www.proteinatlas.org/ENSG00000147894-C9orf72/single+cell> and Tabula Muris: <https://tabula-muris.sf.czbiohub.org/visualizations>), it would be intriguing to explore whether diffuse DPRs are similarly expressed in glial cells in C9orf72 ALS/FTD patients using cryo-sections rather than paraffin-embedded tissue.

8. The authors claim that they studied a mouse model of ALS but used the hippocampus for snRNA-seq analysis. The experiment should be repeated with motor cortex or spinal cord tissues. If it is too much work to redo snRNA-seq, then a better justification for using hippocampus should be provided.

We agree that this is an important point. In this model, neurodegeneration first appears in the hippocampus and is most severe in this region at the endpoint defined by ethical regulations. Similar changes occur in the neocortex and, less pronounced, also in the spinal cord (LaClair et al. Acta Neuropathol 2020). To focus on an area with strong neurodegeneration and facilitate consistent regional dissection, we prioritized the hippocampus for snRNA-seq. This explanation has been incorporated into the revised manuscript. We added bulk RNAseq data from spinal cord of ~7 week old end stage animals from LaClair et al, which confirms cholesterol dysmetabolism (revised Figure 1a) and DOL gene expression (new Fig. 6a), albeit at a slower pace.

9. Seven subtypes of microglia should be identified by single-cell analysis. It is unclear why they clustered microglia into only two groups in their study.

We appreciate the reviewer's observation. While clustering our snRNAseq dataset, we had compared multiple resolutions. Using the clustree package and manual inspection of the UMAPs, we chose a resolution of 0.45 as the best compromise of sensitivity and specificity. These settings, revealed two main microglia states, dividing and non-dividing. While microglia were subclustered into seven (or more) groups in related scRNA-seq studies (e.g. PMID: 39068182), snRNA-seq analysis poorly covers many transcripts associated with microglia activation, which greatly lowers resolution for such analyses (PMID: 32997994). Forced subclustering into seven groups in our dataset would yield diffuse and biologically uninformative clusters. We mention this limitation in the revised discussion.

10. The results from the GA-CaMK2a model should be removed, as they only used one mouse per genotype. Alternatively, if they wish to draw a solid conclusion, they should study additional mice.

We acknowledge the reviewer's concerns and have removed the scRNA-seq data (original Fig. 5e/f and S10-S13) from the GA-Camk2a mouse model due to the limited sample size (n=1). These data were originally included as they supported a non-cell-autonomous mechanism of action and were consistent with snRNA-seq findings from the GA-Nes model. Although this was meant as a validation experiment, we recognize the biological limitations of n=1 for the scRNAseq data. In consultation with the editor, we have removed this dataset, as also requested by Reviewer #3.

To strengthen our conclusions, we have retained and expanded the histological analysis conducted in multiple GA-Camk2a animals (original Fig. 5g, now Fig. 6f). Additionally, we have included new bulk RNAseq data and Serpina3n immunofluorescence, which confirm the induction of DOLs (as shown via key DOL marker Serpina3n) in the GA-Camk2a model driven by neuronal Camk2a-Cre (new Fig. 6a/d/e). Furthermore, we provide evidence of DOL formation in the rNLS8 mouse model, where TDP-43 Δ NLS expression is driven by the neuron-specific neurofilament heavy chain promoter (new Fig. 6b/c). These findings reinforce the role of neuronal aggregates in inducing DOLs and support our overarching hypothesis of non-cell-autonomous mechanisms.

Reviewer #2:

In this manuscript, Rezaei and colleagues describe cholesterol alterations in ALS and focus on the oligodendrocyte population to The study is potentially informative but has major limitations such as the use of a very aggressive and artificial mouse model and reduced translation to human (the RNAseq in Fig. 1 is helpful as an entry point but more human-based data are needed to increase the relevance of the findings).

We thank the review for the helpful comments and provide additional data in two other ALS mouse models as well as human iPSC-derived neurons.

My major points are listed below:

1. How did the authors compare the different RNAseq in Fig. 1? How do they deal with batch effect and heterogeneity of the datasets when integrating them? More information and quality controls must be provided.

We have clarified in the revised manuscript that the heatmap in Figure 1a integrates previously published RNA-seq datasets. For the revised manuscript we added further published and unpublished datasets for validation as requested. In addition, we analyzed new patient data from TargetALS as explained below. These datasets are presented for exploratory purposes without reanalysis of original statistics due to the lack of a common manipulation/genotype. The method section and the legends have been updated to explicitly acknowledge the differences in species and experimental conditions.

2. The human RNAseq from the van Rheen publication has an heterogenous genetic

background, the authors should highlight C9orf72 patients or perform a second layer of analysis with C9-mutant individuals.

We agree with the reviewer’s suggestion. To address this point, we have included a subset analysis for C9orf72 ALS vs controls and C9orf72 ALS vs sporadic ALS patients using an updated cohort TargetALS (see comment 3). While the cholesterol dysmetabolism was found also in C9orf72 ALS patients, our analysis did not reveal C9orf72 specific alterations in the cholesterol pathway. This important new data is shown in the extended Fig 1a.

3. The altered expression of the key genes identified by the authors must be confirmed in independent experiments both in mice and human samples.

We have clarified in the revised text that the gene expression changes presented in Figure 1A were derived from previously published studies. To further strengthen our conclusions, we now included replication studies in additional cohorts of GA-CFP mice and rNLS8 mice. We validated the cholesterol dysmetabolism in GA-Nes in a pseudobulk analysis of the snRNAseq dataset (new Fig S9).

For the human spinal cord data (LaClair et al.), the original analysis utilized a large sample size that cannot be replicated by qPCR. For that study, we analyzed all publicly available spinal cord RNA-seq data from TargetALS as of February 2020, using a linear model accounting for gender and anatomical subregions. To address the reviewer’s concerns, we separately reanalyzed the original dataset from February 2020 (cohort #1: cervical: 13 controls, 85 sporadic ALS, 25 C9orf72 ALS; thoracic: 8 controls, 39 sporadic ALS, 7 C9orf72 ALS; lumbar: 11 controls, 74 sporadic ALS, 19 C9orf72 ALS) and all additional spinal cord samples made available between February 2020 and June 2024 (cohort #2: cervical: 6 controls, 48 sporadic ALS, 3 C9orf72 sporadic ALS; thoracic: 1 control, 7 sporadic ALS; lumbar: 5 controls, 48 sporadic ALS, 5 C9orf72 ALS). Importantly, both cohorts yielded nearly identical results, reinforcing the robustness of the findings (reviewer Figure R1). For simplicity and improve statistical power for the gender comparison, we show only data from the combined cohorts in the manuscript. These new data included in the revised manuscript strengthen the conclusion that cholesterol synthesis is reduced in ALS tissue, while cholesterol export pathways are enhanced.

Reviewer Figure R1 related to Figure 1a: Gene expression changes in the cholesterol pathway from historical bulk RNAseq dataset in GA-Nes (endstage hippocampus, cortex and spinal cord), GA-CFP (thoracic spinal cord, 31 weeks, two independent cohorts) and rNLS8 (hippocampus, after 3 weeks of transgene induction, two independent cohorts) ALS mouse models and patient spinal cord. Log₂ fold changes compared to controls are shown in the heat map. Adjusted p-values from the original analysis are shown, because different genotype and species prevent a combined re-analysis with common RNAseq pipelines. Asterisks indicate significant changes. Upregulation of export pathway genes and downregulation of synthesis genes suggest cholesterol overload.

4. CD treatment should be performed in other ALS models, such as the TDP43 used in for the transcriptome in Fig. 1, but information on other DPR species must also be provided.

We appreciate the reviewer's interest in extending the findings to other ALS models. To address this, we tested CD treatment in the TDP-43 Δ NLS expressing rNLS8 mouse line. Unfortunately, this model can progress even faster than the GA-Nes model. After transgene induction in mature mice (9-10 weeks of age), we had to terminate the mice within 2-3 weeks due to rapid weight loss due to European animal welfare regulations. Daily treatment with CD (2g/kg, s.c.) from the time of transgene induction did not attenuate weight loss, reduce neurofilament levels or improve hindlimb clasping in this model (see Figure R2 below). As this was a single experiment with no opportunity to optimize treatment conditions, we prefer not to include these data in the revised manuscript. For example, the rNLS8 model should have progressed much slower if the transgene was induced earlier (at 5 weeks of age in Walker et al, Acta Neuropathol 2016).

However, bulk RNA-seq and immunofluorescence confirmed the induction of DOLs (new Figure 6a-c) and the cholesterol dysregulation signature was even more pronounced in rNLS8 mice than in the GA-Nes model (extended Figure 1A).

Other ALS models available to us on short notice were not suitable for therapeutic studies: The GA-CFP and GA-Camk2a lines progress too slowly for treatment studies within the timeframe even of an extended revision. Heterogeneity in the survival time in the GA-Camk2a line would also require prohibitively large cohorts for sufficient statistical power. Dichotomous phenotypes in our poly-PR expressing line (PR-Nes, LaClair et al, ANP 2020) with half severely affected mice and half unaffected mice precludes meaningful treatment studies. Nevertheless, we confirmed the expression of DOLs markers in the GA-Camk2a line in the new Figure 6 in n=4 replicates. The *C9orf72* BAC lines available to us show only subtle phenotypes and bulk RNAseq data does not show induction of DOL signature genes. CD has been previously tested in SOD1 mice by others, but was not beneficial (<https://pubmed.ncbi.nlm.nih.gov/37489926/>). Based on these data, we hypothesize that the efficacy of CD may be specific to poly-GA or *C9orf72* pathology, but could potentially extend to other neurodegenerative diseases. This interpretation aligns with the revised discussion.

Reviewer Figure R2: rNLS8 mice treated with CD starting at the time of transgene induction. **A)** rNLS8 mice lose weight after transgene induction, which is not slowed by CD treatment. **B)** Serum NFL levels were not significantly affected by CD treatment. **C)** rNLS8 mice developed hindlimb claspings in the second week after transgene induction. Scoring (0 to 5) revealed no significant difference between CD and vehicle-treated transgenic mice.

5. Since the treatment is more effective in females, analysis of the transcriptome by separating the sexes should additionally be provided to justify these differences. Also, this must be as well compared to human data

We appreciate this insightful comment and acknowledge the limitation of having analyzed only female samples in our snRNA-seq dataset. To address this, we reanalyzed the RNA-seq data from Figure 1a, segregating results by sex. We found no significant differences in the GA-Nes and rNLS8 mice, but our cohort size is not well powered for this subgroup analysis. Therefore, we include that data for the reviewer only.

In contrast, analysis of a large patient cohort revealed a significantly more pronounced cholesterol dysregulation signature in male patients (revised Figure 1a). Moreover, male GA-Nes seem to reach the humane endpoint slightly faster (median 41 vs median 43 days, see Figure 1b/c). Based on these data, we hypothesize that slower disease progression in females may make them more responsive to therapeutic interventions like CD. This interpretation has been incorporated into the revised discussion.

Reviewer Figure R3 related to Figure 1a: Gene expression changes in the cholesterol pathway from historical bulk RNAseq dataset in GA-Nes (combined 5 and 7 week old GA-Nes mice using a model taking age and gender into account) and rNLS8 (hippocampus, after 3 weeks of transgene induction, two independent cohorts) ALS mouse models and patient spinal cord. Log₂ fold changes compared to controls or male vs. female are shown in the heat map. No obvious difference in cholesterol pathways of male vs. female GA-Nes and rNLS8 mice are detected, in contrasted to clear gender effects in the larger human ALS cohort #1 and #1+#2. N see response to question 3.

6. Fig. 1E is not informative in its present form. This might be due to the low quality of the Figure after downloading from the online system but nevertheless the authors must provide higher-resolution imaging (such as STED) must be performed and quantified to better characterize the GA localization, especially because the authors focus later on oligodendrocytes.

We apologize for the suboptimal quality of the originally provided figures, which resulted from the use of a 20x objective without averaging during confocal image acquisition. These images were intended for general visualization rather than colocalization analysis. To address this, we have now performed colocalization analysis using a 63x objective, generating significantly improved, high-resolution images. These much crisper images are included in the revised supplemental Figure S2 and provide clear and reliable data for colocalization analysis without using super-resolution microscopy, which is not available to us. In addition, we replaced Figure 1e with higher-quality overview images of the dentate gyrus, including NeuN co-staining (revised Figure 1f).

7. On the same line, the authors claim based on the SN RNAseq in Fig. 2 that GA expression results in “depletion of astrocytes”, this should be better contextualized in relation to the data in Fig. 1E.

We agree with the reviewer that the description of astrocyte depletion was misleading. The apparent reduction in astrocytes was due to sampling bias in the FANS/snRNA-seq data and not biological depletion, because the absolute number of microglia increased in GA-Nes mice and we analyzed 50% NeuN+ and 50% NeuN- cells using snRNAseq. The language has been corrected in the revised text. We quantified the different glia from histological sections: This showed a 5x increase in Iba1+ microglia, which was not significantly affected by CD

treatment. While oligodendrocytes numbers were not unaffected by poly-GA expression or CD treatment, the number of S100 β positive astrocyte number was even slightly increased in GA-Nes mice, with a trend towards reduction by CD treatment. This new data is shown in the new Figure S6 of the revised manuscript.

8. Fig. 2 b-c-d: I have major concerns on the validity of the PCs used for the analysis. How many PCs were identified in total? I am doubting if PC2 is really informative since there is a dramatic drop of % from PC1 (between 80 and 90% in the 3 PCAs) and PC2 (4 to 7%). This might represent noise in the dataset and must be better characterized. The shift of the samples upon treatment is indeed explained by PC2 and the authors must provide more detailed analysis on this. Euclidian distance should be calculated and PC loading provided to identify the genes responsible for this shift. Do the genes that mostly drive this transcriptional change pass quality control for expression abundance?

We appreciate the reviewer's insightful comments on principal component (PC) analysis and have carefully implemented this excellent suggestion. As suggested in the next comment, PCA analysis has been expanded to include all genes, and the revised analysis is now presented in Figures S7 and 2b-d. The results are very similar to the PCA with the top 500 variable genes, retaining the clear separation of CD and vehicle-treated GA-Nes mice in the oligo 1 and 2 population in PC2. Accordingly, we have replaced the original plot with this more comprehensive analysis.

In the revised PCA, PC1 now explains 63% and 49% of the variability in the Oligo_mat1 and Oligo_mat2 clusters, respectively, while PC2 accounts for 7% and 10%. We have further included the top genes contributing to each principal component, the average expression levels of PC1 and PC2 genes, and the Euclidean distances between groups in a new supplemental Table S2. Indeed, Plin4 is a key contributor to PC2 in the Oligo_mat1 cluster. Moreover, the Euclidean distances between CD and vehicle-treated GA-Nes mice are largest for the Oligo_mat1 cluster, underscoring the impact of CD treatment on oligodendrocyte transcriptional states.

9. As a supplemental information, the same PC analysis must be provided with the entire dataset and not only with the top 500 genes.

As mentioned above, we now show PCA plots based on all gene expression data in the revised figures Fig S7 and Figures 2b-d. The focus on the top 500 genes is the default setting in the plotPCA() function in DESeq2 package, which is probably meant to reduce computation time.

10. Are the genes "rescued" by CD sorted according only to LFC or also adj P value?

We appreciate the opportunity to clarify this point. Genes "rescued" by CD treatment were selected based on an adjusted p-value ($P_{adj} < 0.05$) and a log-fold change (LFC) threshold of ± 1 . We added this important information directly in the revised Fig. 2e/f.

11. The effect of CD on myelination is interesting: does it apply only to the corpus callosum or also to the white matter of the spinal cord? Did the authors observe amelioration in the motor phenotype upon treatment?

We appreciate the interest in our myelination data. As shown in LaClair et al ANP2020, the gene expression changes in different brain regions and the spinal cord are similar, but less severe in the spinal cord. Neurodegeneration starts in the hippocampal CA2 region, while motor neuron loss is only evident at 6-7 weeks of age, coinciding with the humane endpoint required by European animal welfare regulations.

Since motor deficits could not be robustly assessed at P40, we analyzed the transcriptome of the spinal cord at this stage (Figure R4 shown below). The data show only a modest reduction in myelin gene expression in GA-Nes mice, with no significant rescue observed following CD treatment in either males or females. Since the deficit is so modest at this age, we think adding this data to the manuscript is not helpful.

However, we have significantly expanded the myelination data in the corpus callosum through additional EM analysis (new Figure 4d/e and S13a-c. These findings confirm the reduction in myelinated axons in GA-Nes mice and demonstrate a significant rescue by CD treatment. This expanded dataset provides robust evidence for the therapeutic effect of CD on myelination deficits in GA-Nes mice.

Reviewer Figure R4 related to Figure 4b: Expression changes of myelin genes from historical bulk RNAseq dataset in GA-Nes (hippocampus, neocortex and spinal cord) and new bulk RNAseq from CD-treated GA-Nes spinal cord compared to snRNAseq data from oligodendrocyte clusters. GA-Nes mice show a modest but reproducible reduction in myelin genes that is not affected by CD treatment.

12. *Expressing GA only in neurons trigger a similar phenotype, which the authors address as “non-cell autonomous” effect. This is rather a superficial explanation and the mechanism needs to be better elucidated to clarify why suffering neurons impact in this way myelinating cells.*

We have chosen the GA-Camk2a model due to the well-established restricted expression of Camk2a Cre in excitatory neurons (Minichiello et al., Neuron 1999). In response to requests from reviewer 1 and 3, we removed the scRNAseq data from GA-Camk2a mice due to limited sample size (n=1). However, new bulk RNAseq data demonstrate expression of the DOL signature genes in this mouse line, and immunofluorescence confirms the induction of Serpina3n (new Figure 6). Additionally, we show that DOL markers are induced in the rNLS8 model, where TDP-43 Δ NLS expression is largely neuron-specific, driven by the

Neurofilament heavy chain promoter (new Figure 6). These findings collectively hint at a non-cell-autonomous effect of neuronal aggregates on DOL induction. Nevertheless, given the significant expression of poly-GA in oligodendrocytes in the GA-Nes mice used for most of this study, we have revised the manuscript to toned down our claims accordingly.

13. The main cholesterol-related alterations observed in the polyGA mice must be investigated in hiPSC-derived neurons or organoids. Which of the alterations are conserved? Is CD treatment beneficial also in human models? This might also help in addressing the previous point.

We appreciate this insightful suggestion. In the absence of robust human-derived neuron/oligodendrocyte co-culture models, we investigated the direct effects of poly-GA on cholesterol metabolism by overexpressing (GA)₁₄₉-GFP in iPSC-derived human neurons. Poly-GA expression resulted in a minimal but significant reduction in the expression of several cholesterol biosynthetic enzymes compared to GFP controls, without detectable changes in lipid export or storage pathways (2.5 mg/ml) (new Figure S13d). Interestingly, CD treatment significantly upregulated cholesterol biosynthetic pathways in both (GA)₁₄₉-GFP and GFP-expressing cells, indicating that CD depletes the neuronal cholesterol pool and triggers compensatory activation of biosynthesis. These findings align with the minimal alterations observed in cholesterol biosynthesis, export, and storage pathways in the neuronal populations of GA-Nes mice in our snRNAseq dataset (see above and new Figure S9). Together, these results argue against poly-GA-induced excess cholesterol accumulation in neurons, as seen in oligodendrocytes, and highlight distinct cell-type-specific effects of poly-GA pathology on cholesterol metabolism.

Reviewer #3:

Rezaei et al., presents interesting findings on the beneficial effects of the cholesterol binding drug CD in improving survival of female Nestin Cre-poly GA mice – a mouse model for C9ORF72 related ALS/FTD. The authors have used existing transcriptomics data set from various ALS animal models and patient autopsy to set up a hypothesis that that decreased cholesterol biosynthesis. To test this hypothesis, they have administered CD to the poly GA mice which showed reversed expression changes in mature oligodendrocytes and neuroblasts in the hippocampus of young adult transgenic mice that exhibit myelin defects. CD drug has been known to target cholesterol polyesters and there is some evidence in the article to support this with the drug normalising elevated cholesterol polyesters in the midbrain of young adult Nes Cre-poly GA mice. The authors also highlight a cholesterol ester binding protein, Plin4, to be specifically upregulated in oligodendrocytes in transgenic mice with expression partially rescued with CD, advocating for lipid metabolism as a viable therapeutic target. Overall, this study is of relevance in understanding ALS disease pathology, redirecting focus back to oligodendrocytes, a largely understudied cell type. Moreover, the specific effect of the CD drug on oligodendrocyte maturation and subsequent improvement in survival rates in poly GA transgenic mice is noteworthy.

The authors place much importance on the cholesterol biosynthesis pathway being affected in oligodendrocytes leading to myelin defects via a non-cell autonomous mechanism in the CamK2-poly GA mice. Further characterisation is needed to support some aspects of this hypothesis. The manuscript presents limited data on cholesterol synthesis gene expression changes in oligodendrocytes in transgenic mice (with and without CD treatment). The myelin changes in transgenic mice need further studies and the inclusion of CD treated transgenic mice. Moreover, it would be important to elucidate if the poly GA inclusions are present in

oligodendrocytes as well as neurons in their primary Nestin Cre driven mouse model to shed light on disease aetiology and their potential key finding of oligodendrocytes/cholesterol pathway as a target for therapeutics.

We sincerely thank Reviewer #3 for their positive and thoughtful evaluation of our work and the interest in the role of oligodendrocytes and cholesterol in ALS. Based on the insightful comments we refined the manuscript to clearly show that cholesterol dysmetabolism is most severe in oligodendrocytes in GA-Nes mice, which is partially rescued by CD treatment. In addition, we show that CD treatment partially rescues myelin loss in the corpus callosum of GA-Nes mice. Colocalization analysis using endogenous GFP fluorescence reveal a previously unrecognized widespread expression of GFP-(GA)₁₄₉ in oligodendrocytes. Therefore, we further characterizing induction of DOLs in the GA-Camk2a model with pure neuronal expression and additional show evidence for DOL formation in a TDP-43 mouse model. Thank you for your valuable feedback, which has significantly strengthened our study.

Specific comments

1) Fig 1: The drug CD specifically improves survival of female transgenic mice, normalising NfL levels in the blood. Authors should also publish Nfl levels from the male mice as well to shed light on the gender specific role of CD in this mouse model.

We appreciate this suggestion. We repeated the CD treatment in male GA-Nes mice. Consistent with the lack of survival benefit in male mice, NfL levels were not reduced in CD-treated transgenic males. These data are now included in the new Fig. 1e.

2) Fig 1: Although measurement of Nfl levels, as a surrogate for neurodegeneration, is commendable and will be helpful in translational studies, noting the discovery nature of this study the authors should undertake conventional histopathology studies to understand the cellular differences neuronal count, astrocyte/microglial reactivity/oligodendrocytes, to associate which cells contribute to the observed survival benefit.

Thank you for this insightful comment. We have quantified the densities of neurons and glial cells in hippocampal sections using Cresyl Echt Violet staining and immunofluorescence (new Figure S6). Consistent with previous findings, we observed a significant loss of neurons in the CA2 region of the hippocampus in GA-Nes mice and a slight trend toward rescue by CD treatment (Figure S6e). The density of neuroblasts was significantly reduced in GA-Nes mice, and CD treatment did not significantly restore this population (Figure S6d).

In contrast, the number of oligodendrocytes was unaffected by poly-GA expression or CD treatment, indicating that their cellular abundance remains stable (Figure S6a). However, GA-Nes mice showed a dramatic increase in Iba1-positive microglia, consistent with the activation profile observed in our snRNA-seq data (Fig. S5 and S6b) and similar results in our GA-CFP mouse model (Schludi et al, Acta Neuropathol 2017). In addition, the number of astrocytes was significantly increased in GA-Nes mice, and CD treatment showed a slight trend towards normalization.

These findings suggest that CD treatment primarily improves axonal myelination, which may contribute to the reduced NfL levels and observed survival benefit in treated mice.

3) Fig 1: The lifespan experiments have used weight loss as a proxy for humane endpoints in Fig1C. It would be good if the authors could depict the weights as a graph with information

from each animal, in addition to the survival plots since this would be a more accurate representation of the effect of CD on survival of transgenic mice. Moreover, the survival plots show that only about 25-50% of the female mice are significantly improved with CD treatment. The authors should discuss these findings and comment on heterogeneity in the survival, detailed histopathological analysis might help.

We have plotted individual weight trajectories in the new Figure S1 to provide greater transparency. Survival heterogeneity likely reflects intrinsic variability in disease progression in this model, which is a common feature of ALS mouse studies. This has been discussed in the revised manuscript.

4) Fig 1: The authors allude towards poly GA inclusions being present in both neurons and glia, however, data to support this claim is missing in the current manuscript. They could consider including immunohistochemistry for glial cell type markers co-labelled with CA-GFP with subsequent quantifications to understand if aggregate pathology is also originating in oligodendrocytes and driving myelin defects directly via a cell-autonomous mechanism.

We agree with the reviewer that a clearer characterization of cell types expressing poly-GA in the GA-Nes model is critical. Thus, we have analyzed co-localization of endogenous GFP-(GA)149 fluorescence with glial markers (new Figure S2). Our analysis shows colocalization of Iba1-positive microglia with small poly-GA dots, which may reflect either phagocytic uptake or low levels of endogenous expression. By contrast, most of S100 β -positive astrocytes exhibit high levels of poly-GA expression and most of oligodendrocytes also show small areas of poly-GA expression. These findings highlight a previously underappreciated widespread glial expression of the transgene in the GA-Nes model, which was not apparent to this extent using immunohistochemistry or RNAscope (LaClair et al, Acta Neuropath 2020).

Since these findings argue for a cell-autonomous component of poly-GA in oligodendrocytes of GA-Nes mice, we confirmed cholesterol dysmetabolism in the GA-Camk2a mice (new Figure 6a) and a second cohort of GA-CFP mice (extended Figure 1a), both of which exhibit pure neuronal poly-GA expression. Moreover, we confirmed induction of key DOL markers in GA-Camk2a mice and the newly added rNLS8 mouse model with predominant neuronal transgene expression driven by the neurofilament heavy chain promoter (new Figure 6b-f).

Finally, given the widespread expression of C9orf72 in neurons and glia (as shown in the Human Protein Atlas: <https://www.proteinatlas.org/ENSG00000147894-C9orf72/single+cell> and Tabula Muris: <https://tabula-muris.sf.czbiohub.org/visualizations>), it would be intriguing to explore whether diffuse DPRs are similarly expressed in glial cells in C9orf72 ALS/FTD patients using cryo-sections rather than paraffin-embedded tissue.

5) Fig 2: the authors present elegant snRNAseq data from different cell types showing that mature oligodendrocytes and neuroblasts have the most transcriptional changes with CD treatment normalising several key myelin genes. However, there is no specific dataset examining the status of cholesterol biosynthesis genes in each cell type. Given that the foundation of the study examines the cholesterol pathway, the authors should consider including this dataset, particularly in oligodendrocytes. The authors also remark on the relative depletion of astrocytes, neuroblasts and oligodendrocytes in the transgenic mice from the snRNAseq data. Given the non-empirical nature of FANS/snRNAseq, it would be good to support these claims with immunohistochemistry for cell type specific markers in the transgenic/CD treated mice.

We appreciate the reviewer's interest in cholesterol biosynthesis pathways. We have generated cell-type-specific expression profiles for cholesterol-related genes from the snRNA-seq data focusing on the main neuronal and glial populations for clarity (new Figure S9). This analysis revealed shows that cholesterol export and storage pathways were mostly induced in oligodendrocytes of GA-Nes mice and less consistently in microglia and astrocytes. CD treatment had the greatest rescue effect on oligodendrocytes.

We agree with the reviewer that the description of astrocyte depletion was misleading. The apparent reduction in astrocytes was due to sampling bias in the FANS/snRNA-seq data and not biological depletion, because the absolute number of microglia increased in GA-Nes mice and we analyzed 50% NeuN+ and 50% NeuN- cells using snRNAseq. The language has been corrected in the revised text. We quantified the different glia from histological sections: This showed a 5x increase in Iba1+ microglia, which was not significantly affected by CD treatment. While oligodendrocytes numbers were not unaffected by poly-GA expression or CD treatment, the number of S100 β positive astrocyte number was even slightly increased in GA-Nes mice, with a trend towards reduction by CD treatment. This new data is shown in the new Figure S6 of the revised manuscript.

6) Fig 3D: For the Serpin A3N immunoblotting experiment the authors have used pooled lysates from number of brains. This masks the biological variability, thus, authors should consider repeating this experiment keeping animals separate with subsequent quantifications to reflect experimental variability and robustness of the data.

We have repeated the Serpina3n immunoblotting with samples from individual mice, and the quantitative analysis confirmed a significant reduction of SerpinA3N in CD-treated GA-Nes mice. This important result is shown in the new Fig 3e of the revised manuscript.

7) Given the marked gene expression changes in oligodendrocytes, the authors briefly examine myelin changes in transgenic mice. This dataset is limited and potentially needs further characterisation to assess if myelination is impaired in transgenic mice or if it is a type of axonopathy that is observed. The authors could include quantifications such as G-ratio and/or axon diameter measurements. The images look like there is a drastic loss of myelinated axons in the corpus callosum. How does this compare with the hippocampus wherein the snRNAseq was performed? In general, the manuscript examines several regions in different experiments (midbrain, hindbrain, hippocampus, and cortex/corpus callosum). It would be worth discussing region specific changes as well as maintaining consistency within brain regions for gene expression and immunoblotting/immunohistochemistry.

Thank you for raising this important point. We have performed electron microscopy (EM) to assess myelination in the corpus callosum in a new cohort of CD-treated GA-Nes mice. Preliminary analyses suggested that the primary difference in GA-Nes mice was a reduction in the number of myelinated axons rather than thinning of the remaining myelin sheaths. Therefore, we did not analyze G-ratios in all animals. Quantitative analysis of myelination in the corpus callosum confirmed a significant loss of myelinated axons in GA-Nes mice. Importantly, this deficit was significantly rescued by CD treatment (new Fig. 4d/e). These findings were further validated in an independent earlier cohort using Luxol Fast Blue (LFB) staining shown below (Figure R5).

Reviewer Figure R5 related to Figure 4d/e: Luxol fast blue staining show myelin loss in GA-Nes mice. We quantified thickness of the corpus callosum above the CA2 region of the hippocampus for better comparability. This revealed partial rescue by CD, consistent with data from an independent cohort analyzed by EM (Figure 4). N=3 in each condition, representing independent biological replicates.

We agree with the reviewer that it would be more elegant to perform all assays in the hippocampus. However, we utilized different brain areas to maximize the use of available tissue, as poly-GA is uniformly expressed throughout the CNS (LaClair et al, ANP 2020). Since neuron loss is most prominent in the hippocampus, we used the area of corpus callosum that is associated with hippocampus to investigate myelination. Additionally, ethical guidelines mandated by the German government, including adherence to the 3R principles (Replacement, Reduction, Refinement), necessitated careful and efficient use of animals. Furthermore, ongoing construction near our facility has significantly impacted breeding efficiency over the past two years, which posed additional logistical challenges. We had to setup 35 matings (15 of which were unsuccessful) to generate 94 experimental animals for the revision. We mention the reasons for this limitation and the uniform poly-GA pathology in the revised manuscript.

8) Myelin analysis in CD treated mice is missing and would be key to understanding if the drug reverses only oligodendrocyte gene expression or if this is also translated to oligodendrocyte function being restored. It could be helpful to perform analysis at earlier timepoints to better understand if the myelin phenotypes observed is due to developmental myelination or demyelination. For eg. at P21 when drug is administered, is myelin comparable in the genotypes as this would speak towards a rescue of demyelination by CD.

As described above, CD-treatment significantly restores the density for myelinated axons in the corpus callosum (extended Fig 4d/e), indicating that CD supports oligodendrocyte function in GA-Nes mice.

We agree that distinguishing between poly-GA-induced demyelination and impaired developmental myelination is important. To address this, we quantified the density of myelinated axons at postnatal day 21 (P21), prior to treatment initiation. Myelination in GA-

Nes mice and control littermates was comparable at this stage (new Fig. S13b/c), hinting that that active demyelination occurs between P21 and P40 and is ameliorated by CD treatment.

However, since the density of myelinated axons significantly increases in WT mice between P21 and P40 (and to a lesser extent in transgenic mice) we cannot entirely exclude the possibility of poly-GA-dependent inhibition of developmental myelination during this period. This nuance has been acknowledged and discussed in the revised manuscript.

9) The final section of the manuscript examines the expression of the cholesterol specific gene Plin4. The specific upregulation of Plin4 in only mature oligodendrocytes is striking and interesting. If the authors could examine Plin4 expression in human postmortem ALS tissue or in a human stem cell model this would be very helpful in understanding the importance of Plin4/lipid metabolism as a therapeutic target in ALS human pathology.

We appreciate the reviewer's suggestion to validate Plin4 expression in human postmortem ALS tissue. Preliminary analyses indicate increased Plin4 expression in a subset of C9orf72 ALS patients, but significant variability within the control group limits the robustness of these findings. Addressing this variability will require a larger cohort with better-matched cases and controls using optimized staining techniques. Due to these constraints, we are unable to provide conclusive data on this point in the timeframe of a revision.

Instead, we provide some human data in iPSC neurons (new Figure S13d), which shows minimal reduction in cholesterol biosynthetic enzyme expression upon poly-GA expression in neurons, but compensatory activation of biosynthesis upon chronic CD treatment. Together, these results argue against poly-GA-induced excess cholesterol accumulation in neurons, which was seen in oligodendrocytes. These results further highlight distinct cell-type-specific effects of poly-GA pathology on cholesterol metabolism.

10) The Camk2a-Cre data alludes largely towards unpublished data from another manuscript in preparation. The data included in this manuscript is quite preliminary with limited evidence supporting the conclusions. Moreover, the scRNAseq experiment is underpowered with only one biological replicate. The authors should consider including further characterisation of the Camk2aCre model, particularly in relation to CD treatment or completely removing this dataset and soften the discussion around non-cell autonomous role. This dataset is quite crucial as it can address if poly GA pathology and subsequent oligodendrocyte dysfunction occurs via cell-autonomous or non-cell autonomous mechanisms.

We acknowledge the reviewer's concerns and have removed the scRNA-seq data (original Fig. 5e/f and S10-S13) from the GA-Camk2a mouse model due to the limited sample size (n=1). These data were originally included as they supported a non-cell-autonomous mechanism of action and were consistent with snRNA-seq findings from the GA-Nes model. Although this was meant as a validation experiment, we recognize the biological limitations of n=1 for the scRNAseq data. In consultation with the editor, we have removed this dataset, as also requested by Reviewer #1.

To strengthen our conclusions, we have retained and expanded the histological analysis conducted in multiple GA-Camk2a animals (original Fig. 5g, now Fig. 6f). Additionally, we have included new bulk RNAseq data and Serpina3n immunofluorescence, which confirm the induction of DOLs in the GA-Camk2a model driven by neuronal Camk2a-Cre (new Fig.

6a/d/e). Furthermore, we provide evidence of DOL formation in the rNLS8 mouse model, where TDP-43 Δ NLS expression is driven by the neuron-specific neurofilament heavy chain promoter (new Fig. 6b/c). These findings collectively hint at a non-cell-autonomous effect of neuronal aggregates on DOL induction. Nevertheless, given the significant expression of poly-GA in oligodendrocytes in the GA-Nes mice used for most of this study, we have revised the manuscript to toned down our claims accordingly.

11) As a more general comment, the authors should include additional details in methods/figure legends mentioning number of litters and if error bars are values from individual animals/sections.

Thank you for your practical suggestion. We have updated the Methods and figure legends to include details on error bars, which represent variability among individual animals unless stated otherwise. The number of litters and replicates for each experiment has also been specified.

Point by point response to referees

Reviewer #3 (Remarks to the Author):

The authors have addressed most of the key issues raised previously.

Few minor suggestions,

1) It is unclear how the extended survival in a subset of female mice is achieved. Authors observed CD treatment only increased myelination and did not alter other cell types. Did the authors perform pathological characterisation only in female mice or both sexes were included? It would be necessary to clarify in the text. If the study combines both sexes, this suggests improvement in myelination alone is not sufficient to increase survival and perhaps other mechanisms are involved. Would suggest to discuss this.

We appreciate the constructive feedback. We have now explicitly mentioned in the result section that we after observing lifespan extension and reduction of NfL exclusively in female mice, we continued all further downstream experiments in female mice only. It should now be clearer that after figure 1, all figures containing GA-Nes mice are solely focused on female mice. While an explanation of the gender-specific response on a molecular level falls outside the scope of our study, we discuss some possible causes: “Our gender-specific responses are consistent with data in NPC mice although the mechanism remains elusive³⁹. Interestingly, cholesterol dysmetabolism is more pronounced in male ALS patients and testosterone is affecting myelination in mice³⁸. The promyelinating effects of testosterone and differential cholesterol metabolism between the two genders may explain the dichotomy we observed in response to CD treatment, although further studies are needed”.

Lifespan extension in a subset of female mice was consistent in multiple cohorts, however, an explanation regarding why only a subset of females benefit from CD eludes us because at the humane endpoint the pathological hallmarks are similar, and at an earlier timepoint it is unclear which mice would benefit from the life expansion benefit of CD. We hypothesize that individual differences in hormone regulation, metabolism, ADME variability of CD, epigenetic differences, and adiposity might play a role, but further studies outside the scope of our study are needed for any definitive conclusions.

2) Human iPSC neuron study is not fully characterised, ie identity of neurons, efficiency of poly-GA formation, suggest remove this data as it doesn't add further knowledge to the study.

In agreement with the editor, we prefer to keep the data in the manuscript, because it was specifically requested by the reviewers in the first revision.

To further support the characterization of our human iPSC-derived neurons, we provided additional immunofluorescence images demonstrating (GA)₁₄₉-GFP and GFP expression, consistent with our previous characterization of lentiviral vectors in rat primary neurons (May et al., Acta Neuropath. 2014). These data confirm efficient poly-GA expression in the human iPSC-derived neurons (new Figure S13d).

The differentiation of iNeurons was performed using NGN2 induction at the NPC stage, a well-established and widely used protocol. Our prior studies (Czuppa et al. Cell Reports 2022, Strauß et al., Front. Cell Dev. Biol. 2021) demonstrated robust neuronal identity in this model, with MAP2 as a pan-neuronal marker and BRN2 as a marker for upper cortical layer neurons. In addition, we have uploaded RNA sequencing data to GEO (GSE285209), providing a comprehensive transcriptional profile of these neurons. We have modified the text accordingly in the revised manuscript.